# Mitochondrial signatures shape phenotype switching and apoptosis in response to PLK1 inhibitors

Émilie Lavallée, Maëline Roulet-Matton, Viviane Giang, Roxana Cardona Hurtado, Dominic Chaput(iD), Simon-Pierre Gravel(iD)

**PLK1 inhibitors are emerging anticancer agents that are being tested as monotherapy and combination therapies for various cancers. Although PLK1 inhibition in experimental models has shown potent antitumor effects, translation to the clinic has been hampered by low antitumor activity and tumor relapse. Here, we report the identification of mitochondrial protein signatures that determine the sensitivity to approaches targeting PLK1 in human melanoma cell lines. In response to PLK1 inhibition or gene silencing, resistant cells adopt a pro-inflammatory and dedifferentiated phenotype, whereas sensitive cells undergo apoptosis. Mitochondrial DNA depletion and silencing of the ABCD1 transporter sensitize cells to PLK1 inhibition and attenuate the associated pro-inflammatory response. We also found that nonselective inhibitors of the p90 ribosomal S6 kinase (RSK) exert their antiproliferative and pro-inflammatory effects via PLK1 inhibition. Specific inhibition of RSK, on the other hand, is anti-inflammatory and promotes a program of antigen presentation. This study reveals the overlooked effects of PLK1 on phenotype switching and suggests that mitochondrial precision medicine can help improve the response to targeted therapies.**

## Introduction

Melanoma is a skin cancer that originates from melanocytes, which are specialized epidermal cells responsible for melanin deposition and protection from UV rays. While early lesions are easily managed, late-stage cancer poses significant challenges for treatment owing to its aggressive nature and propensity to metastasize. Melanoma is responsible for over 80% of skin cancer deaths, and the 5-yr survival rate of advanced melanoma (stage IV) is less than 30% (Saginala et al, 2021). Mutations in the BRAF kinase are observed in ~40–60% of melanoma patients, and other frequent melanoma subtypes involve activating mutations in RAS isoforms (20–30% of patients) or loss of NF1 (10–15%) (Gutierrez-Castaneda et al, 2020; Vanni et al, 2020; Randic et al, 2021). BRAF mutations lead to constitutive activation of the pro-tumorigenic mitogen-activated protein kinase cascade (BRAF-MEK-ERK) (Alqathama, 2020; Castellani et al, 2023). Despite the improvement of progression-free survival in patients treated with BRAF inhibitors in combination with MEK inhibitors, the emergence of drug resistance through a variety of mechanisms remains a major obstacle in implementing sustainable responses and improving patient outcomes.

The p90 ribosomal S6 kinase (RSK) family contains four members, RSK1-4, that share ~80% homology. RSK isoforms have context-specific expression patterns and exert diverse functions by phosphorylating many protein substrates (reviewed in Houles et al [2018]). Specifically, RSK1/2 were shown to be up-regulated in melanoma cells and to act as predominant targets of ERK in melanoma (Old et al, 2009; Galan et al, 2014; Wu et al, 2022). RSK1/2 were also shown to mediate resistance to chemotherapy and BRAF inhibition (Ray-David et al, 2013; Wu et al, 2022). Recent reports have proposed that RSK1/2 are potential therapeutic targets in melanoma with constitutive MAPK activation to overcome resistance to BRAF and MEK inhibitors (Kosnopfel et al, 2017, 2023). Despite the importance of these findings and the identification of new drug candidates, it is crucial to consider the selectivity of RSK inhibitors because some off-target effects may contribute to the observed antitumoral activity. One notorious unselective RSK inhibitor, BI-1870, was shown to bind to the active site of dozens of unrelated kinases such as Polo-like kinase (PLK) 1/3, aurora kinase B, and PIK3 ([Neise et al, 2013; Aronchik et al, 2014; Edgar et al, 2014; Roffe et al, 2015] and reviewed in Sun et al [2023] and Wright and Lannigan [2023]). For other promising compounds with good biodistribution, tolerability, and specific antitumor activities, data on selectivity are often incomplete or unpublished, which may be detrimental to the identification of molecular targets and understanding their mechanism of action. Thus, we hypothesized that the antiproliferative effects of several RSK inhibitors might be caused by off-target inhibition of other protein kinases, such as PLK1, a master regulator of cell division and candidate for targeted therapy with increased expression in melanoma (Strebhardt et al, 2000; Kneisel et al, 2002; Chiappa et al, 2022). This possibility prompted us to revisit the real impact of selective RSK inhibition on melanoma proliferation and answer outstanding questions, such as the effects

---

Faculté de Pharmacie, Université de Montréal, Montréal, Canada

Correspondence: sp.gravel@umontreal.ca

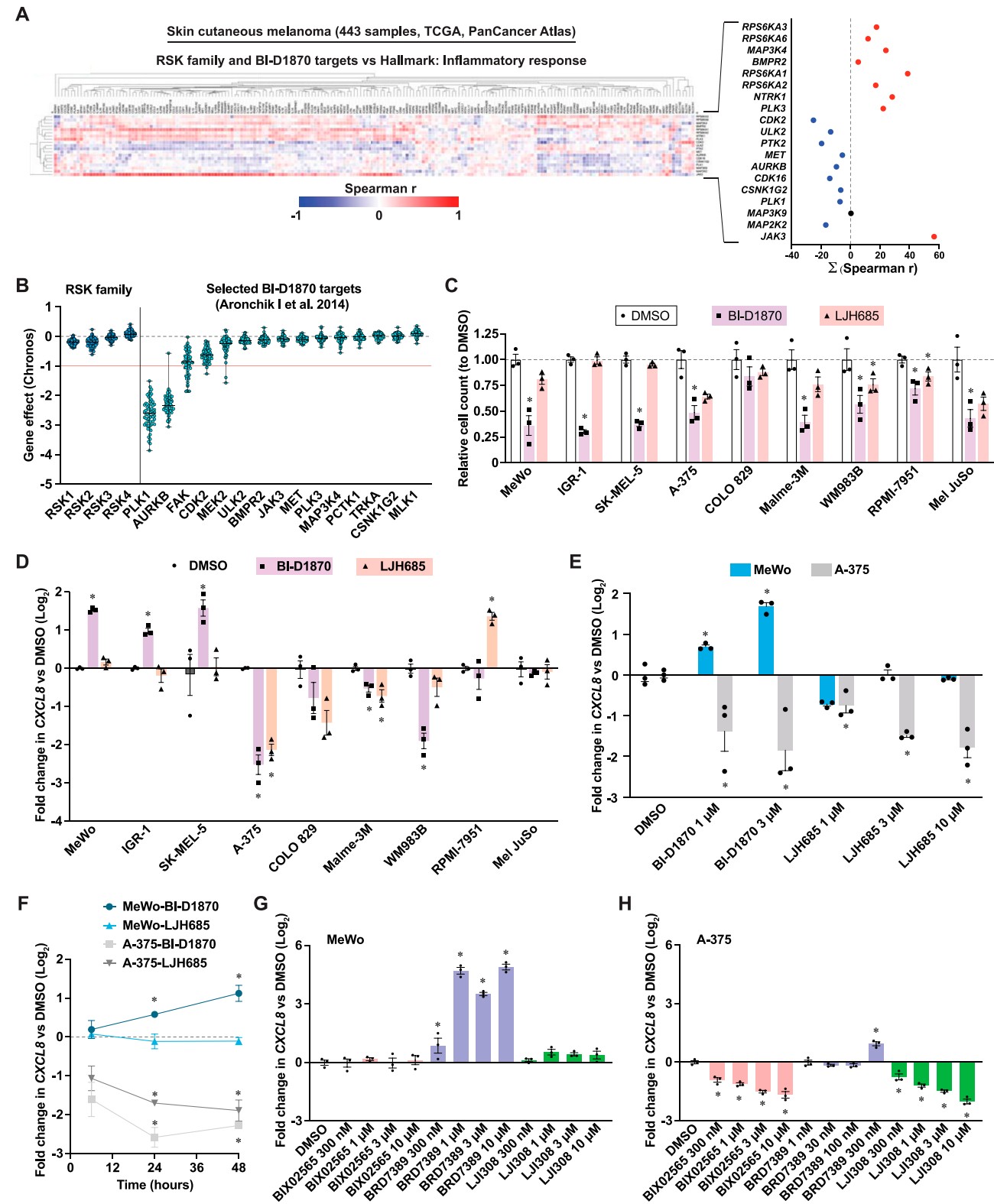

**Figure 1. The antiproliferative and pro-inflammatory effects of RSK inhibitors in melanoma cells are inhibitor- and cell line-specific.**
See also Fig S1. **(A)** Correlation of transcripts encoding for RSK family members and putative targets of BI-D1870 with inflammatory response transcripts (Hallmarks, MSigDB). The right graph plots the summation of all Spearman correlations shown the heat-map on the left. The following publicly available dataset was used: Skin Cutaneous Melanoma (TCGA, PanCancer Atlas). **(B)** Impact of CRISPR-Cas9-mediated loss of genes encoding for RSK family members and putative targets of BI-D1870 on proliferation dynamics (Chronos algorithm) of human melanoma cell lines. The red line (–1) is the median of all common essential gene scores. Each dot represents a

on tumor inflammation and phenotype switching. Our systematic investigation of RSK inhibitors with distinct specificity profiles allowed us to decipher RSK-specific roles in inflammation and antigen presentation. Our results show that PLK1 is a putative target of unselective RSK inhibitors and mediates their antiproliferative and pro-inflammatory effects. In addition, they indicate that the pronounced heterogenous response observed under RSK and PLK1 inhibition is shaped by the expression of mitochondrial proteins associated with a resistance signature.

# Results

### The antiproliferative and pro-inflammatory effects of RSK inhibitors in melanoma cells are inhibitor- and cell line-specific

To examine the association between RSK expression and inflammation in human melanoma, we performed correlation analyses using the skin cutaneous melanoma (SKCM) RNA-seq data set of human melanoma tumors from the TCGA PanCancer Atlas. We also analyzed transcripts encoding putative off-targets of BI-D1870, a notorious nonselective RSK inhibitor (reviewed in Wright and Lannigan [2023]). These analyses revealed that all RSK family members correlate positively with inflammation-associated transcripts (Fig 1A), whereas several BI-D1870 targets, such as PLK1, CDK2, and PTK2, showed a negative correlation. Next, the proliferative effect of knockout and gene silencing of the same kinases on melanoma cell lines was examined using the DepMap portal. Knockout and gene silencing of RSK family members had an overall minimal or insignificant effect on the proliferation of melanoma cell lines, whereas the targeting of PLK1 had a dramatic effect, followed by the targeting of Aurora B, FAK, and CDK2 (Figs 1B and S1A). These analyses suggest that the antiproliferative and pro-inflammatory effects of RSK inhibitors, such as BI-D1870, BRD7389, PMD-026, AF007, and fisetin in previous melanoma studies (Salhi et al, 2015; Kosnopfel et al, 2017, 2023; Theodosakis et al, 2017; Sechi et al, 2018; Li et al, 2022) could be due to the unselective targeting of other kinases, such as PLK1 (Fig S1B).

To examine the antiproliferative and pro-inflammatory effects of two RSK inhibitors, we treated a panel of nine human melanoma cell lines with the unselective BI-D1870 (Edgar et al, 2014) and the selective LJH685 (Aronchik et al, 2014). BI-D1870 had significant antiproliferative effects on 8/9 cell lines, whereas LJH685 had little to no effect (Fig 1C). To study the impact of these inhibitors on inflammation, we measured the transcript levels of interleukin 8 (IL-8, gene CXCL8), a chemokine linked to melanoma progression, phenotype switching and resistance (reviewed in Filimon et al [2021]). Strikingly, the responses to BI-D1870 and LJH685 appeared to be compound- and cell line-specific (Fig 1D). BI-D1870 had a pro-inflammatory effect on three melanoma cell lines, MeWo, SK-MEL-5, and IGR-1, whereas LJH685 had no effect on these cell

lines or even had anti-inflammatory effects in A-375, COLO 829, Malme-3M, and WM983B cells. In view of their clearly opposite responses to inhibitors, we selected the MeWo and A-375 cell lines for subsequent experiments. At the dose used of 3 $\mu M$, the effect of the drugs on viability was minimal in both cell lines (Fig S1C and D). We also observed a dose-dependent effect of these drugs on the induction and repression of CXCL8 in MeWo and A-375 cells, respectively (Fig 1E). Since CXCL8 levels were measured 72 h after stimulation, we investigated whether the anti-inflammatory effect of RSK inhibitors was due to negative feedback. In cells treated for shorter periods (6, 24, and 48 h), RSK inhibition with BI-D1870 and LJH685 resulted in reduced levels of CXCL8 in A-375 cells after 6 h, while in MeWo cells, an increase in CXCL8 was observed after 24 h with BI-D1870 only (Fig 1F). To further strengthen these observations, we examined the effect of additional RSK inhibitors on CXCL8 in MeWo and A-375 cells. Treatment with the unselective RSK inhibitor BRD7389 (Fomina-Yadlin et al, 2010) also strongly induced CXCL8 expression in MeWo cells (Fig 1G). Interestingly, the pro-inflammatory effects of BRD7389 could not be observed in A-375 above 300 nM due to cell death (Fig 1H). Two additional selective RSK inhibitors, BIX02565 (Edgar et al, 2014) and LJI308 (Aronchik et al, 2014), had no effect on MeWo cells but had strong dose-dependent anti-inflammatory effects in A-375 cells (Fig 1G and H). Taken together, these results reveal an important heterogeneity of response to RSK inhibitors and suggest that targets other than RSK, such as PLK1, may be responsible for the induction of a pro-inflammatory response or apoptosis in a cell type-dependent manner.

### Nonselective RSK inhibition induces a pro-inflammatory gene expression program linked to phenotype switching

To further characterize the antiproliferative and pro-inflammatory effects of the nonselective RSK inhibitor BI-D1870, we performed RNA-seq analyses in MeWo cells treated for 3 d with this compound. As expected, a detailed proliferation analysis of MeWo cells shows that BI-D1870 leads to rapid and sustained proliferation arrest (Fig 2A). Next, we performed gene functional classification of differentially expressed transcripts (Fig 2B) using the Reactome collection of gene sets on g:Profiler. Up-regulated transcripts were linked to pathways such as neutrophil degranulation, innate immune system, and cellular stress responses (Figs 2C and S2A), whereas down-regulated transcripts were almost exclusively linked to cell cycle pathways (Figs 2D and S2B). Gene set enrichment analyses (GSEA) of normalized reads using the Hallmarks collection of gene sets confirmed the induction of an inflammatory response and the repression of cell cycle genes linked to E2F and checkpoints (Figs 2E and S2C). Interestingly, Cytoscape network representation of GSEA using the GO:BP collection of gene sets revealed a strong clustering of genes within an immune response cluster that evokes cytoskeletal changes and cell migration (Fig 2F). Since the

---

single cell line; data are shown as median ± min/max. The following publicly available dataset was used: DepMap Portal, CRISPR (DepMap Public 24Q2+Score, Chronos) for the indicated genes. **(C)** Impact of RSK inhibitors on the proliferation of human melanoma cell lines. Cells were treated for 72 h with DMSO 0.1% or RSK inhibitors at 3 $\mu M$. **(D)** CXCL8 transcript levels in human melanoma cell lines treated for 72 h with DMSO 0.1% or RSK inhibitors at 3 $\mu M$. **(E)** CXCL8 transcript levels in human melanoma cell lines treated with escalating doses of RSK inhibitors for 72 h. **(F)** CXCL8 transcript levels in human melanoma cell lines treated with RSK inhibitors for 6, 24 and 48 h. **(G, H)** CXCL8 transcript levels in MeWo and A-375 cells in response to escalating doses of additional RSK inhibitors. Data are shown as mean ± SEM of three independent experiments. *P < 0.05, one-way ANOVA with Dunnett's multiple comparison test.

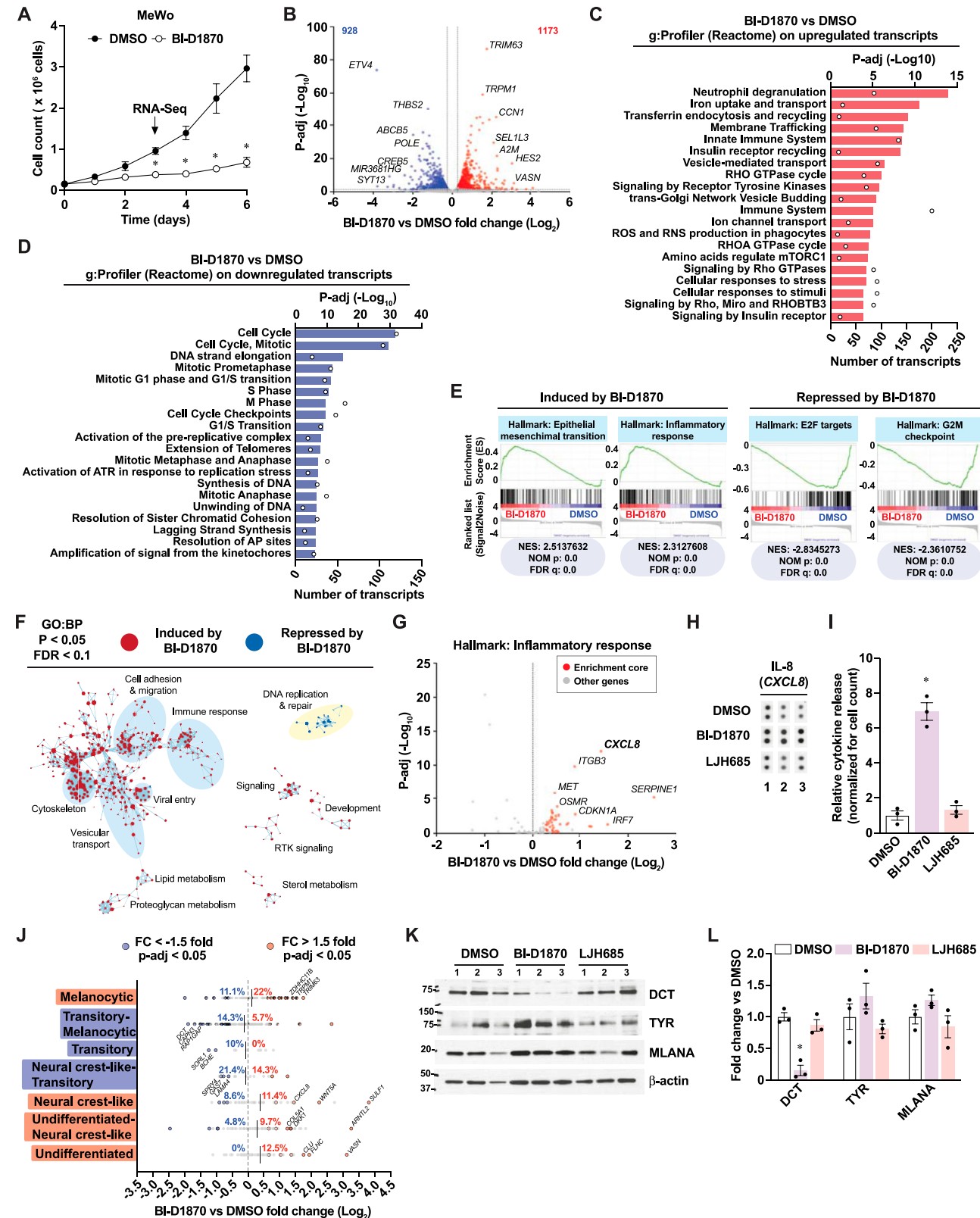

**Figure 2. The unselective RSK inhibitor BI-D1870 induces a pro-inflammatory response linked to phenotype switching.**
See also Fig S2. **(A)** Proliferation curve in MeWo cells treated with DMSO 0.1% or BI-D1870 (3 μM) for 72 h. **(B)** Differentially expressed transcripts in MeWo cells treated with BI-D1870 for 72 h compared to DMSO. Numbers in upper corners indicate the number of up-regulated (red) and down-regulated (blue) significant transcripts based on threshold values of fold change > 1.2 and p-adj < 0.05. **(C, D)** Functional gene analysis on differentially expressed transcripts in MeWo cells treated with BI-D1870 for 72 h

inflammation marker *CXCL8* was among the most enriched up-regulated transcripts from the Inflammatory Response gene set (Fig 2G), we next asked whether the corresponding cytokine IL-8 was secreted by MeWo cells after treatment with BI-D1870. The analysis of cytokines from conditioned media of MeWo cells revealed a sevenfold induction of IL-8 (Fig 2H and I) and significant induction of other cytokines associated with melanoma progression and therapeutic resistance, such as GROα (*CXCL1*) (Dhawan & Richmond, 2002; Botton et al, 2011) and IL-6 (Soler et al, 2023) (Fig S2D and E). Since proliferative and inflammatory statuses are tightly associated with phenotype switching in melanoma (reviewed in Hossain and Eccles [2023]), we examined the impact of BI-D1870 on the expression of transcripts previously associated with seven phenotypic states (from differentiated/melanocytic to undifferentiated) (Tsoi et al, 2018). Strikingly, BI-D1870 had a complex impact on these transcripts, leading to a polarization toward both the melanocytic stage and neural crest-like/undifferentiated states and a depletion of transitory stages (Fig 2J). Treatment of MeWo cells with BI-D1870 reduced the expression of Dopachrome Tautomerase/TRP-2 (*DCT*), a modulator of the melanocytic state (Lenggenhager et al, 2014; Muccioli et al, 2023), while treatment with LJH685 had no impact (Fig 2K and L). Taken together, these results suggest that nonselective RSK inhibitors, such as BI-D1870, can influence melanoma phenotype switching and the tumor immune landscape.

### Selective RSK inhibition is associated with anti-inflammatory gene expression and immunometabolic rewiring

To examine the anti-inflammatory effects of selective RSK inhibition, we performed RNA-seq analyses on A-375 cells treated with LJH685. Treated cells proliferated as control cells at 72 h and over a 6-d period (Fig 3A). Notably, the pro-inflammatory transcript *CXCL8* was among the top differentially expressed transcripts in A-375 cells (Fig 3B). Functional gene classification of differentially expressed transcripts in these cells revealed the enrichment of diverse pathways linked to extracellular matrix remodeling, signaling, and metabolism (Fig 3C and D). Furthermore, characterization of down-regulated metabolic transcripts allowed us to identify 197 transcripts implicated in diverse metabolic pathways, such as the metabolism of lipids, carbohydrates, amino acids, cofactors, and nucleotides (Fig 3E). Since the anti-proliferative effects of LJH685 are negligible in A-375 cells, these results suggest that RSK inhibition reorganizes the metabolic landscape to bolster proliferation. GSEA further revealed a reduction in gene sets related to hypoxia and inflammation (Figs 3F and S3A). Strikingly, visualization of enriched GO:BP gene set networks revealed that LJH865

led to global repression of immune response pathways, except for the induction of pathways associated with antigen processing and presentation (Fig 3G). A detailed volcano plot representation of these genes shows a significant induction of several MHC class I and class II transcripts in A-375 cells treated with LJH865 (Fig 3H). Other RSK-selective inhibitors, such as BIX02565 and LJI308, were also able to induce antigen presentation genes, such as *HLA-DQA2*, *CD74*, and *HLA-DOA* (Figs 3I and J and S3B). Importantly, transcripts associated with antigen presentation were also induced in MeWo cells treated with BI-D1870 (Fig S2A), reinforcing a RSK-specific role. We next asked whether selective RSK inhibition could sensitize cells to IFN-γ, a well-established inducer of antigen presentation genes in melanoma (Ghosh et al, 1989). The induction of antigen presentation transcripts was observed in A-375 cells treated with LJH685 or IFN-γ, and the combination of both treatments resulted in stronger induction (Fig 3K). This suggests that RSK activity status can likely modulate basal and IFN-γ-induced antigen presentation in melanoma tumors. As performed for the study of BI-D1870 in MeWo cells (Fig 2J), we examined the impact of LJH685 on the expression level of transcript signatures associated with seven differentiation stages in melanoma. This RNA-seq analysis shows that RSK markedly lowers markers of undifferentiated states while increasing neural crest-like and transitory markers (including *DCT*) (Fig 3L). Taken together, these results do not support the antiproliferative and pro-inflammatory effects of specific RSK inhibition. Conversely, RSK inhibition appears to reorganize the immunometabolic landscape of melanoma cells by reducing inflammation and rewiring metabolic pathways.

### The pro-inflammatory effects of PLK1 targeting are linked to therapeutic resistance and phenotype switching in a cell line-dependent manner

Unselective RSK inhibitors, such as BI-D1870, were shown to bind to the ATP-binding site of several kinases, such as PLK1 (Aronchik et al, 2014). As suggested in Figs 1A and B and S1A, targeting the expression of these kinases could affect the proliferation and pro-inflammatory response of melanoma cells and tumors. Thus, we hypothesized that the pro-inflammatory effects of BI-D1870 (Fig 1D–F and 2G) and likely BRD7389 (Fig 1G) are not due to RSK inhibition, but likely through another kinase (Fig S1B). To confirm this hypothesis, we examined the induction of *CXCL8* in MeWo and A-375 cells treated for 72 h with inhibitors targeting PLK1/3, CDK2, FAK, ULK1/2, AURORA/B, CDK16, MET, and JAK3, which are kinases known to bind BI-D1870 (Aronchik et al, 2014). While inhibition of ULK1/2 with MRT68921 had similar pro-inflammatory effects in both cell

---

compared to DMSO. **(E)** Significantly enriched gene sets (Hallmarks, MSigDB) in MeWo cells treated with BI-D1870 for 72 h compared to DMSO (GSEA). NES, normalized enrichment score; NOM p, nominal *P* value; FDR, false-discovery rate q value. **(F)** Cytoscape visualization of significantly enriched gene sets in MeWo cells treated with BI-D1870 for 72 h compared to DMSO. **(G)** Volcano plot of the inflammatory response gene set (Hallmarks, MSigDB) from RNA-seq analyses performed on MeWo cells treated with BI-D1870 for 72 h compared to DMSO. Enrichment core (red dots) indicates genes from the GSEA leading edge. **(H)** IL-8 detection by cytokine array in conditioned media from MeWo cells treated with DMSO 0.1%, BI-D1870 (3 μM), or LJH685 (3 μM) for 72 h. Numbers indicate independent experiments. **(I)** Densitometric analysis of three independent cytokine arrays performed in MeWo cells treated with RSK inhibitors for 72 h. **(J)** Modulation of transcripts associated with phenotype switching from RNA-seq analyses performed on MeWo cells treated with BI-D1870 for 72 h compared to DMSO. Percentages indicate the proportion of significantly up-regulated or down-regulated transcripts in each gene set associated with a specific differentiation stage. **(K)** Immunoblotting of differentiation markers in cell extracts from MeWo cells treated with DMSO or RSK inhibitors for 72 h. Numbers above pictures indicate paired independent experiments. **(L)** Densitometric analysis of immunoblotting from MeWo cells treated with DMSO or RSK inhibitors for 72 h. Data are shown as mean ± SEM of three independent experiments. **(A)**: *P* < 0.05, paired *t* test. **(I, L)**: *P* < 0.05, one-way ANOVA with Dunnett's multiple comparison test.

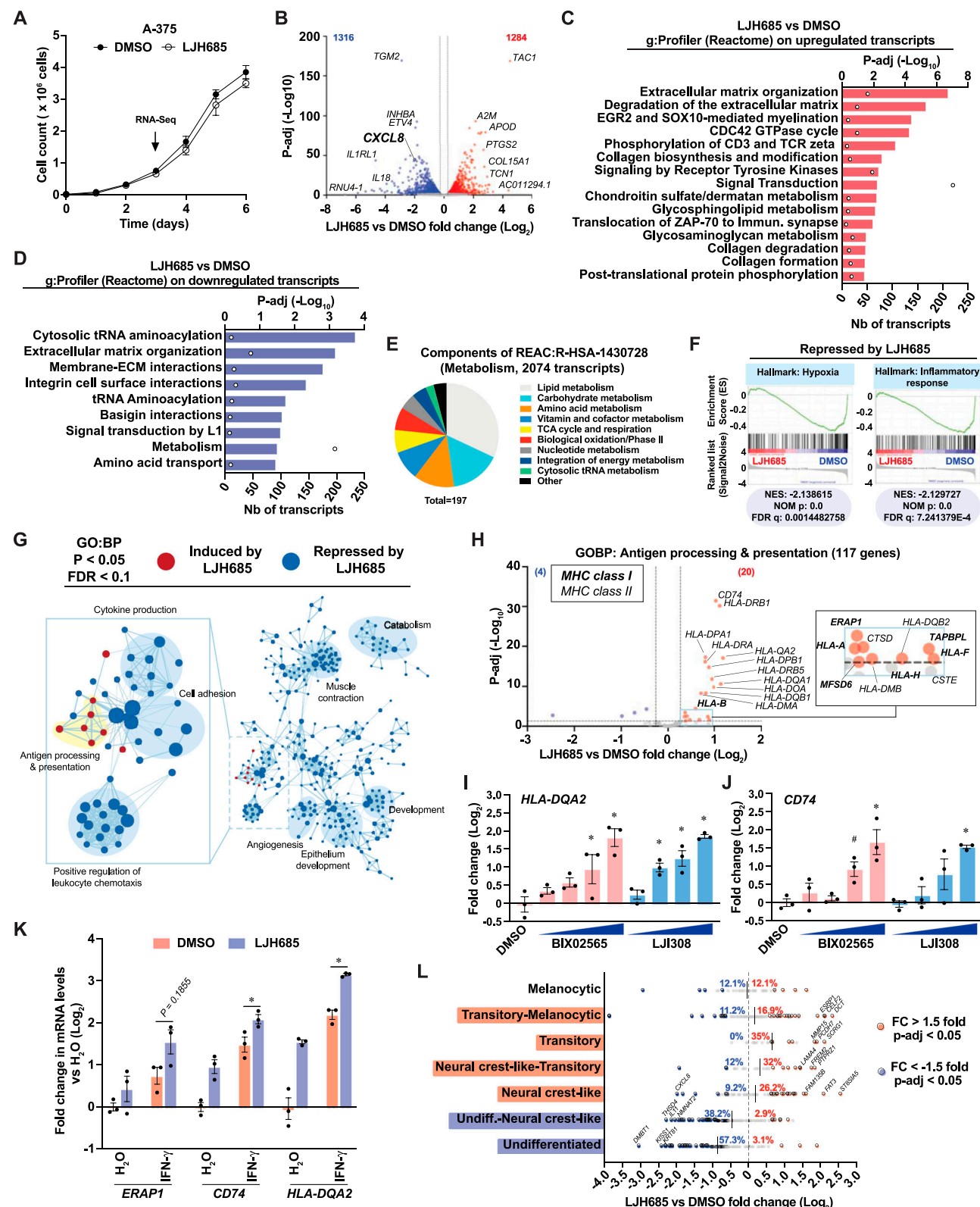

**Figure 3. The selective RSK inhibitor LJH685 induces an anti-inflammatory gene expression program associated with enriched antigen presentation pathway transcripts.**

See also Fig S3. **(A)** Proliferation curve in A-375 cells treated with DMSO 0.1% or LJH685 (3 µM) for 72 h. **(B)** Differentially expressed transcripts in A-375 cells treated with LJH685 for 72 h compared to DMSO. Numbers in the upper corners indicate the number of up-regulated (red) and down-regulated (blue) significant transcripts based on

lines, all other inhibitors had distinct effects (Fig 4A), as observed for BI-D1870 and BRD7389 (Fig 1D and G). The selective PLK1/3 inhibitor BI 6727 (volasertib) (Rudolph et al, 2009) was the most potent inducer of *CXCL8* in MeWo cells, with a half-maximal induction at doses as low as 10 nM. In contrast to the weak effects of the RSK-selective inhibitor LJH685 on cell proliferation and survival (Figs 3A and S1D), BI 6727 induced cell death in A-375 cells with an IC$_{50}$ of 14.1 nM (Fig 4B). Although MeWo cells have an IC$_{50}$ similar to that of A-375 cells (25.4 nM), they show a survival plateau at 30–50% of maximal viability at higher doses of BI 6727, as seen with PrestoBlue HS and crystal violet assays (Fig 4B–D). In response to BI 6727, PARP cleavage and *β*-actin degradation were higher in A-375 cells than in MeWo cells (Fig 4E–G), suggesting that cell death in A-375 cells is related to the induction of apoptosis. As an alternative approach, we silenced PLK1 in MeWo and A-375 cells (Fig 4H). As observed for PLK1 inhibition with BI 6727, PLK1 knockdown (KD) specifically induced *CXCL8* in MeWo cells but had no effect on A-375 cells (Fig 4I). In addition, PLK1 KD in A-375 cells induced PARP cleavage and *β*-actin degradation, supporting proapoptotic effects (Fig 4J and K).

Next, we investigated whether PLK1 targeting could modulate the expression of differentiation markers, as observed for the treatment with BI-D1870 (Fig 2J–L). In MeWo cells, PLK1 inhibition repressed the levels of differentiation markers such as *TYR*, *TYRP1*, and *DCT* (Fig 5A–C). The expression of the DCT protein was also fully suppressed by treatment with BI 6727 (Fig 5D), and PLK1 gene silencing recapitulated these effects (Fig 5E). To further characterize the impact of PLK1 inhibition on phenotype switching, we used RNA-seq data from MeWo cells treated with BI-D1870 (Fig 2) and identified transcripts associated with the proliferative/melanocytic and invasive/dedifferentiated states (Fig 5F). Testing these transcripts revealed that selective PLK1 inhibition with BI 6727 in MeWo cells induces a switch toward a more invasive/dedifferentiated state (Fig 5G), which was also observed with the nonselective RSK inhibitor BRD7389 (Fig 5H). These experiments confirm our hypothesis that nonselective RSK inhibition induces pro-inflammatory effects and dedifferentiation by inhibiting other kinases such as PLK1. These results also suggest that the cell line-specific capacity to mount a pro-inflammatory response to PLK1 inhibition is intimately linked to drug resistance and/or protection from apoptosis.

### Mitochondrial determinants shape the sensitivity and inflammatory response to RSK and PLK1 inhibition

To determine whether a transcriptional signature is associated with the capacity of cells to resist PLK1 inhibition and mount a pro-inflammatory response, we selected melanoma cell lines based on

their *CXCL8* induction pattern in response to BI-D1870 treatment (Fig 1D) and performed GSEA on the corresponding publicly available RNA-seq dataset for each cell line (Fig 6A). These analyses revealed that pro-inflammatory cell lines (MeWo, IGR-1, and SK-MEL-5) were significantly enriched in gene sets related to oxidative phosphorylation (OXPHOS), lipid metabolism, and mTORC1 signaling (Fig 6A and B). In contrast, anti-inflammatory cell lines (A-375, COLO 829, Malme-3M, and WM983B) showed enrichment in gene sets such as EMT, inflammation, and apoptosis (Fig 6A and C). These analyses led us to hypothesize that the response to RSK and PLK1 inhibitors is defined by mitochondria, which is in line with the central role of these organelles in tumor progression and therapeutic resistance in melanoma (reviewed in Kumar et al [2021] and Du et al [2023]). However, we did not find any difference that could match the two groups of cells for mitochondrial DNA (mtDNA) content (Fig 6D) and expression of labile respiratory complex subunits (Fig S4A), although some cell-specific differences were detected. Bioenergetic measurements were similar between the two groups, but indicated that the A-375 cell line exhibited basal and stress-induced characteristics distinct from those of other cell lines. Interestingly, A-375 cells had a higher ratio between oxygen consumption and extracellular acidification rates (Fig 6E) and a higher fraction of mitochondrial respiration dedicated to ATP synthesis (Fig 6F). These results indicate that this cell line efficiently uses its mitochondria under basal conditions, which does not support basal respiration as a modulator of response to RSK or PLK1 inhibition. However, A-375 cells were less efficient at increasing mitochondrial respiration in response to chemical uncoupling (Fig 6G and H) and were more efficient at increasing glycolysis in response to monensin, a Na$^+$ ionophore that maximizes ATP hydrolysis by the Na$^+$/K$^+$-ATPase (Fig 6I and J). These results suggest that A-375 cells have constitutive mitochondrial configurations that make these organelles maladapted to cellular stress, which could be linked to their propensity to engage apoptosis compared with other cells. To explore the potential role of mitochondria in shaping the pro-inflammatory and resistant phenotype of MeWo cells, we used the Rho-0 protocol to generate mitochondria-incompetent cells and tested their response to PLK1 inhibition. Compared with parental cells, Rho-0 cells were characterized by a dramatic reduction in the mtDNA/genomic DNA ratio, reduced expression of labile respiratory proteins subunits, and absence of mitochondrial respiration (Figs 6K and S4B and D). In addition, Rho-0 cells showed a compensatory increase in glycolysis (Figs 6K and S4C). Strikingly, Rho-0 cells showed major changes in their sensitivity profile to BI 6727 treatment compared with MeWo cells

---

threshold values of fold change > 1.2 and p-adj < 0.05. **(C, D)** Functional gene analysis (g: Profiler; Reactome) of differentially expressed transcripts in A-375 cells treated with LJH685 for 72 h compared to DMSO. **(E)** Pie chart associating a functional category (g:Profiler) to 197 metabolic transcripts that are repressed in A-375 cells treated with LJH685 for 72 h compared to DMSO. **(F)** Significantly repressed gene sets (Hallmarks, MSigDB) in A-375 cells treated with LJH685 compared to DMSO (GSEA). NES, normalized enrichment score; NOM p, nominal *P* value; FDR, false-discovery rate q value. **(G)** Cytoscape visualization of significantly enriched gene sets in A-375 cells treated with LJH685 for 72 h compared to DMSO. **(H)** Volcano plot of the antigen processing and presentation transcripts (Hallmarks, MSigDB) in A-375 cells treated with LJH865 for 72 h compared to DMSO. **(I, J)** *HLA-DQA2* and *CD74* transcript levels in A-375 cells treated with DMSO 0.1%, BIX02565, or LJI308 (both at 0.3, 1, 3, 10 *μ*M) for 72 h. **(K)** Antigen processing and presentation gene expression in A-375 cells treated with DMSO 0.1% or LJH685 (10 *μ*M) for 72 h and then treated with human recombinant IFN-γ (1 ng/ml) or water for 24 h. **(L)** Modulation of transcripts associated with phenotype switching from RNA-seq analyses perfomed on A-375 cells treated with LJH685 for 72 h compared to DMSO. Percentages indicate the proportion of significantly up-regulated or down-regulated transcripts in each gene set associated with a specific differentiation stage. Data are shown as mean ± SEM of three independent experiments. **(I, J)**: *P < 0.05; #P < 0.1, one-way ANOVA with Dunnett's multiple comparison test (versus the DMSO control). **(K)**: *P < 0.05, paired *t* test between IFN-γ treatments for a given transcript.

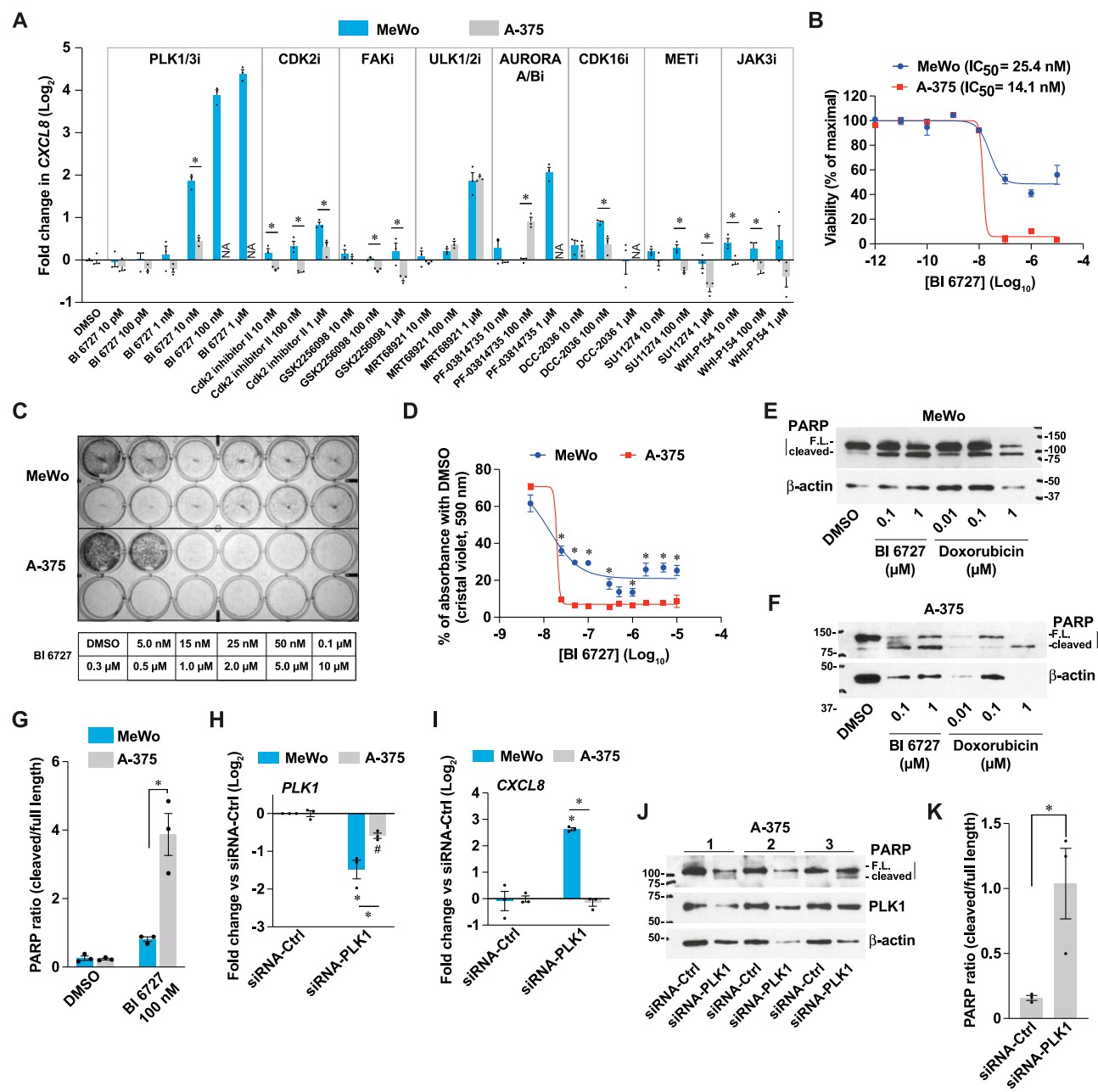

**Figure 4. The pro-inflammatory effects of PLK1 targeting are linked to therapeutic resistance in a cell line-dependent manner.**
**(A)** *CXCL8* transcript levels in MeWo and A-375 cells treated for 72 h with DMSO 0.1% or a panel of inhibitors targeting putative BI-D1870 targets. NA: nonavailable due to cell death. **(B)** Viability of melanoma cells assessed with PrestoBlue HS, expressed as % of maximal viability. **(C)** Representative crystal violet staining of cells treated for 72 h. **(D)** Quantification of crystal violet staining from three independent experiments. **(E, F)** Immunoblotting of cell extracts from cells treated as indicated for 72 h. FL, full-length. **(G)** Densitometric analysis of immunoblotting from three independent experiments. **(H, I)** *PLK1* and *CXCL8* transcript levels in cells 72 h after PLK1 KD. **(J)** Immunoblotting of cell extracts from cells treated for 72 h. Numbers indicate independent experiments. **(K)** Densitometric analysis of immunoblotting from three independent experiments performed with A-375 cells treated for 72 h. Data are shown as mean ± SEM of three independent experiments. **(A)**: *$P < 0.05$, unpaired $t$ test (cell lines were tested separately). **(D, K)**: *$P < 0.05$, paired $t$ test. **(G, H, I)**: *$P < 0.05$, two-way ANOVA with Tukey's multiple comparison test.

(Fig 6L). While being less apoptotic in response to BI 6727 at lower doses (Fig S4E and F), Rho-0 cells lost their resistance to apoptosis at 10 $\mu$M BI 6727 (Fig 6L and M), a dose from the survival plateau observed in MeWo cells (Fig 4B and D). Rho-0 cells were also less potent at inducing *CXCL8* and *IL6* in response to RSK

and PLK1 inhibitors (Fig 6N and O). Taken together, these results suggest that specific expression patterns of mitochondrial proteins may be responsible for the sensitivity and inflammatory profiles of melanoma cells in response to RSK and PLK1 inhibitors.

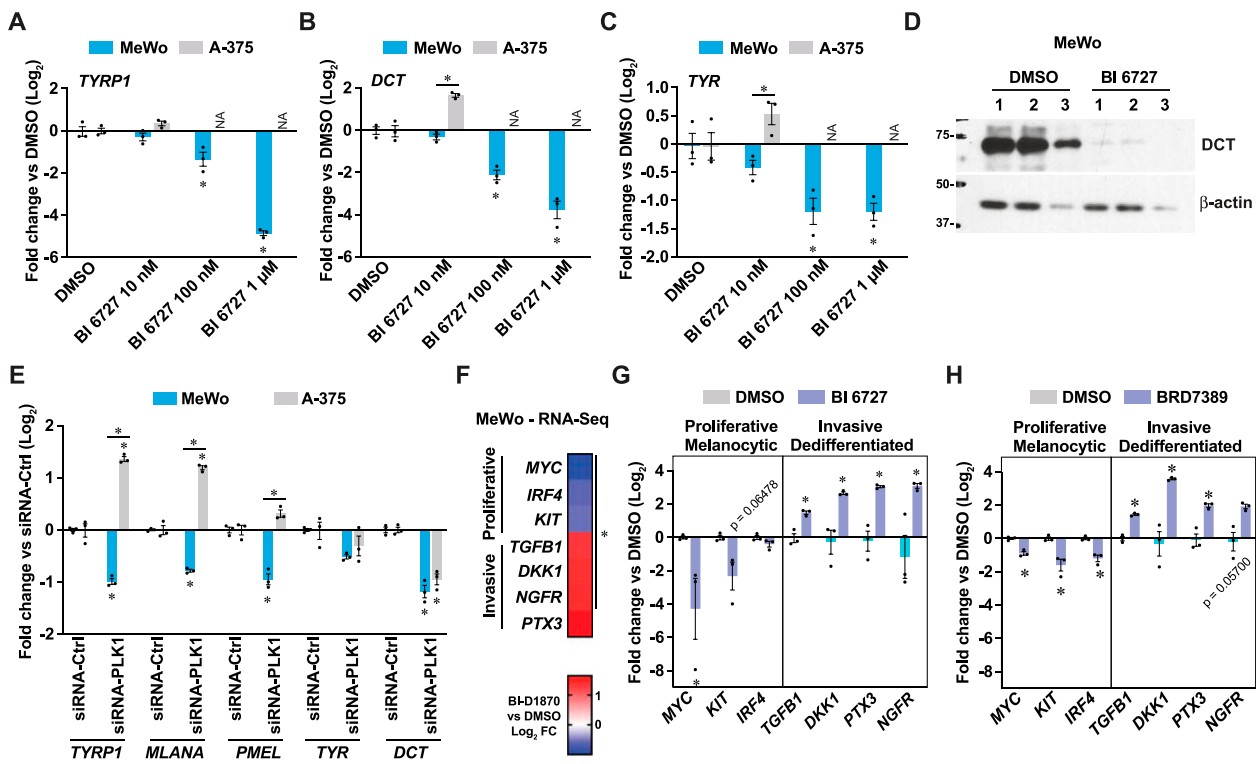

**Figure 5. Resistance to PLK1 inhibition is associated with phenotype switching towards a dedifferentiated state in melanoma cells.**
**(A, B, C)** Gene expression in MeWo and A-375 cells treated for 72 h as indicated. NA, not available due to cell death. **(D)** Immunoblotting of protein extracts from MeWo cells treated for 72 h. Numbers indicate independent experiments. **(E)** Gene expression of differentiation markers in cells treated with siRNA against PLK1 for 72 h compared to non-targeting siRNA. **(F)** Heat-map of Log₂ fold change of specific transcripts linked to phenotype switching in MeWo cells treated with BI-D1870 (3 μM) for 72 h. Derived from RNA-seq analyses in Fig 2. **(G, H)** Gene expression in MeWo cells treated with BI 6727 (1 μM) or BRD7389 (10 μM) for 72 h. Data are shown as mean ± SEM of three independent experiments. **(A, B, C, E)**: *P < 0.05, two-way ANOVA with Tukey's multiple comparison test. **(F)**: * p-adj < 0.05. **(G, H)**: * < 0.05, paired t test.

## Defining actionable mitochondrial signatures associated with resistance to PLK1 targeting in melanoma

To characterize the mitochondrial protein expression patterns associated with resistance to PLK1 inhibition in melanoma cell lines, we performed correlation analyses between transcript expression levels (CCLE) and the impact of PLK1-targeting approaches on cell proliferation (DepMap). PLK1-targeting approaches included gene knockout by CRISPR/Cas9, gene silencing by RNA interference, and PLK1 inhibitors: BI 6727, BI 2536, GSK-461364, HMN-214, NMS-1286937, and GW-843682X (Fig 7A). For the selection of correlating transcripts with good confidence, we chose transcripts that showed a significant correlation with proliferation impact for at least two PLK1-targeting approaches. This led to the primary identification of 881 and 1,501 transcripts that correlate positively and negatively, respectively, with resistance to PLK1-targeting (Fig 7B). Gene functional classification of "sensitizing" transcripts (negative correlation coefficient ρ) revealed enrichment for translation, ribosome biogenesis, and cell division pathways, while "resistance" transcripts (positive ρ) were enriched for development, metabolic, and pigmentation pathways (Fig 7C). Considering that the A-375 cell line is more proliferative than the MeWo cell line (Fig S5A), these analyses suggest that the sensitivity to PLK1 targeting is higher in cell lines that are more committed to cell division, which is expected given

the role of PLK1 in this process. However, these analyses also indicate that metabolic pathways are associated with resistance to PLK1 targeting. In line with our hypothesis that mitochondrial determinants are linked to this resistance, we next identified 124 transcripts encoding mitochondrial proteins that significantly correlate with the sensitivity to PLK1 targeting in melanoma cells. By matching each gene with the gene essentiality (CRISPR/Cas9) data from the DepMap portal, we found that a major fraction of "sensitizing" and "resistance" genes are classified as nonessential (Fig 7D), which opens the possibility of exploiting the targeting of these genes for experimental and therapeutic purposes. All six PLK1 inhibitors had a highly similar correlation pattern with most genes from the mitochondrial signature (Fig 7E). Interestingly, the unselective RSK inhibitor BI-D1870 clustered with both PLK1 KD and PLK1 inhibitors, further supporting the inhibition of PLK1 by this compound. Functional classification of the gene signature, as listed by MitoPathways3.0, revealed that sensitizing transcripts were associated with mitochondrial central dogma (replication, transcription, translation), carbohydrate metabolism, and apoptosis, whereas resistance transcripts were associated with lipid metabolism, amino acid metabolism, and detoxification (Fig 7F). Using STRING, we also performed a protein-protein interaction analysis of the whole mitochondrial transcript signature. This analysis shows that most proteins encoded by the signature are part of a tight interaction network that parallels functional

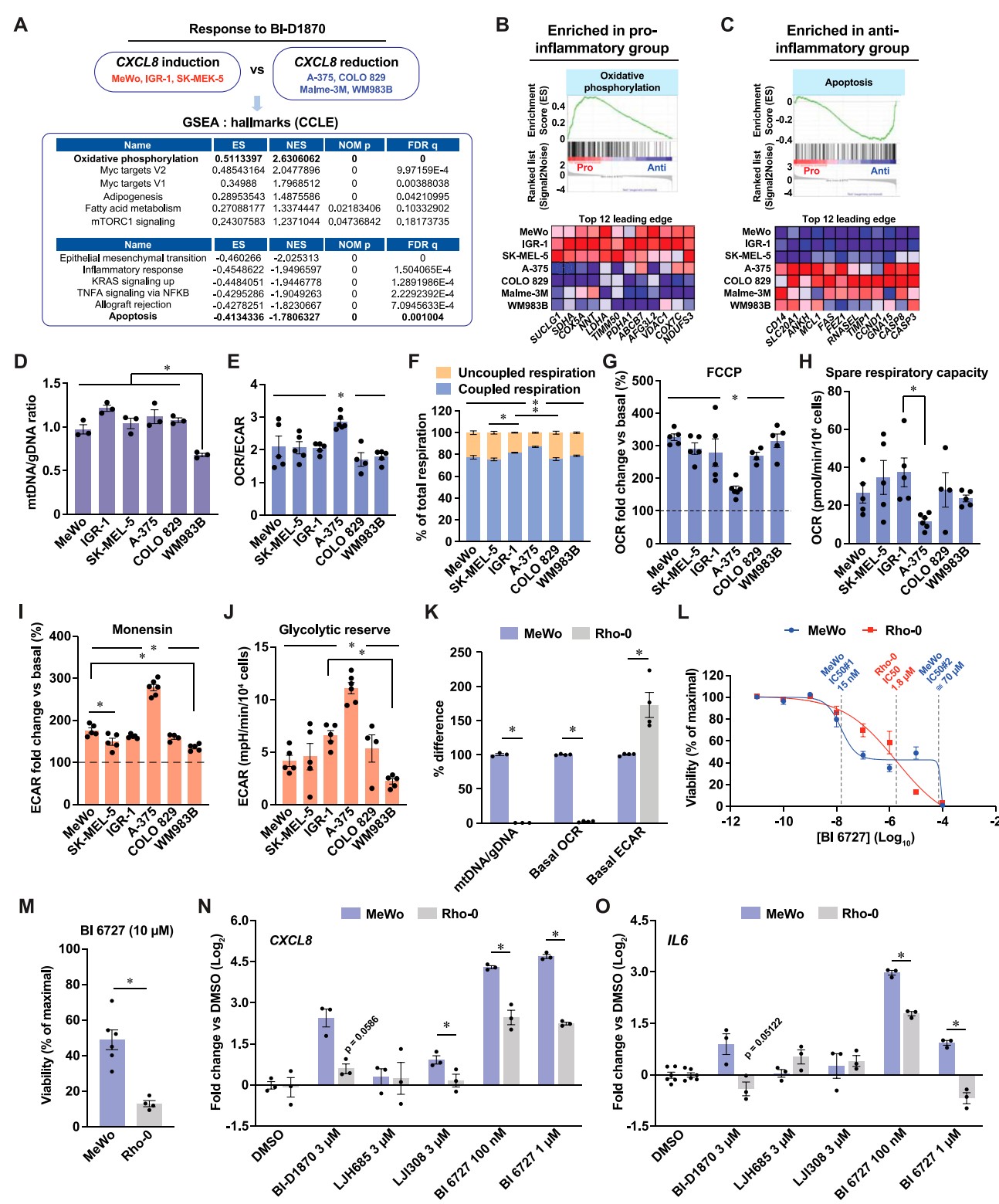

**Figure 6. Mitochondrial determinants shape the sensitivity and inflammatory response to RSK and PLK1 inhibition.**

See also Fig S4. **(A)** Top gene set enrichments identified by GSEA in transcriptomic datasets of human melanoma cells subsets defined by their response to BI-D1870. ES, enrichment score; NES, normalized enrichment score; NOM p, nominal *P* value; FDR q, false-discovery rate q value. The following publicly available dataset was used: Cancer cell line encyclopedia (Broad, 2019). **(B, C)** GSEA and top 12 leading edge transcripts in groups of cells defined by their response to BI-D1870. Pro/anti: cell group with pro- or anti-inflammatory response to BI-D1870. The following publicly available dataset was used: Cancer Cell Line Encyclopedia (Broad, 2019). **(D)** Mitochondrial DNA to genomic DNA ratio in human melanoma cell lines. **(E)** Oxygen consumption rate to extracellular acidification rate ratio in human melanoma cell lines. **(F)** Metabolic organization in melanoma cell lines. **(G)** Maximal induction of Oxygen consumption rate by acute FCCP treatment relative to basal conditions in human melanoma cell

classification analyses, such as mitochondrial central dogma, mitochondrial metabolism, OXPHOS, and apoptosis (Fig 7G). To test whether the targeting of resistance-associated transcripts can indeed affect the response to PLK1 inhibition, we silenced three of these transcripts in MeWo cells, *ABCD1* (ATP-binding cassette subfamily member 1), *BCL2A1* (Bcl-2-related protein A1), and *PRKACA* (cAMP-dependent protein kinase catalytic subunit α), chosen for their strong correlation with inflammatory transcript clusters in human melanoma tumors (Fig S5B–F). Individual KD of these three transcripts reduced the induction of *CXCL8* and *IL6* by BI 6727 treatment, with siRNA targeting *ABCD1* being the most potent treatment (Fig 7H). *ABCD1* KD also lowered the resistance plateau previously observed in MeWo cells treated with high doses of BI 6727 (Fig 7I). Taken together, these analyses show that mitochondrial signatures are associated with distinct responses to RSK and PLK1 inhibitors (Fig 7J) and that targeting specific mitochondrial proteins is a valid approach to fine-tune the sensitivity of melanoma cells to PLK1 inhibition.

# Discussion

The first evidence of PLK1 as a putative target for melanoma treatment emerged from seminal studies showing increased PLK1 expression in melanoma compared with normal tissue and melanocytes. Lower survival of patients with high PLK1 tumoral expression also suggested a prognostic value for this kinase (Strebhardt et al, 2000; Kneisel et al, 2002). The specific induction of apoptosis in melanoma cells versus melanocytes (Schmit et al, 2009) has reinforced interest in targeting PLK1 with small molecules (BI 2536 [Jalili et al, 2011; de Oliveira et al, 2012], BI 6727 [Cholewa et al, 2017]) and launching clinical trials (BI 6727 [Lin et al, 2014] and other inhibitors, reviewed in Chiappa et al [2022]). Over the past few years, PLK1 inhibition has emerged as an attractive option for melanoma treatment in combination with MEK inhibition (Posch et al, 2015; Kohtamaki et al, 2022; Yu et al, 2022), NOTCH inhibition (Su et al, 2021), or in the case of resistance to directed therapies (BRAF and MEK inhibition [Sanchez et al, 2019]). PLK1 inhibitors appear to be versatile anticancer agents for a variety of cancers, and their recent interplay with immunotherapies (Reda et al, 2022; Zhang et al, 2022) will likely open interesting research avenues for melanoma treatment. While PLK1 inhibition in experimental models shows potent antitumoral effects in vitro and in vivo, translation to the clinic has been curbed by modest antitumor activity, tumor relapse, and frequent lack of response (Frost et al, 2012; Stadler et al, 2014; Ellis et al, 2015; Awad et al, 2017; Craig et al, 2022). Few studies have examined the factors that confer resistance to PLK1 inhibition. Mutations in the ATP-binding site of the *PLK1* gene (Burkard et al, 2012; Adachi et al, 2017), increased expression of AXL/

TWIST1/MDR1 (Adachi et al, 2017; Solanes-Casado et al, 2021), and the induction of mitotic slippage at high doses of PLK1 inhibitors (Raab et al, 2015) are among the resistance mechanisms observed after the establishment of resistant cell lines. Herein, we show that some melanoma cell lines, such as MeWo, possess intrinsic resistance towards PLK1 inhibition, without the need to establish resistant cell lines. In response to PLK1 inhibition, MeWo cells become non-proliferative and mount a potent pro-inflammatory response. To our knowledge, this study is the first to highlight the role of mitochondria in establishing cell-intrinsic resistance to PLK1 inhibition.

The role of mitochondria in cancer has long been overlooked due to the propensity of cancer cells to divert cellular bioenergetics towards aerobic glycolysis (Warburg effect). This simplistic view of mitochondria shaded the multiple roles of these organelles in many cellular processes beyond oxygen consumption and ATP synthesis (reviewed in Ruocco et al [2019]). Importantly, mitochondria can potently shape the response to therapies through various mechanisms (Zhang et al, 2016; Du et al, 2023; Mahmood et al, 2024). Here, we found that mitochondria shape the response of melanoma cells to PLK1 inhibition by acting as a switch between immunologically silent cell death and pro-inflammatory survival. We have identified subsets of nuclear genes that encode mitochondrial proteins that can either enhance or decrease the impact of PLK1-directed treatments (pharmacological inhibition, gene silencing, gene knockout) on melanoma cell proliferation. Recently, a genome-wide CRISPR screen allowed the identification of genes that modulate sensitivity to PLK1 inhibition, among which genes involved in chromosome attachment and segregation on the mitotic spindle were identified as sensitizers (Normandin et al, 2023). We examined the data from this study to determine whether our candidate genes for the mitochondrial resistance signature were included. Interestingly, the knockout of the resistance-associated gene *PRKACA* sensitized cells to PLK1 inhibition, while the knockout of 15 "sensitivity" genes from our signature rescued cell proliferation. Although this study used the nonmelanoma cell line NALM-6, these findings suggest that mitochondria are general modulators of response to PLK1 inhibition. *PRKACA* encodes the catalytic subunit α of PKA, a protein kinase that plays pleiotropic roles in metabolism, inflammation, and apoptosis. In melanoma, PKA can modulate BRAF and CRAF (Marquette et al, 2011; Li et al, 2013), and overexpression of PKA induces resistance to MAPK inhibition (Johannessen et al, 2013). PKA has several substrates on the outer mitochondrial membrane and within mitochondria and thus can modulate processes such as OXPHOS, apoptosis, mitochondrial dynamics, and mitophagy (review in Ould Amer and Hebert-Chatelain [2018]). Although PKA appears to be a candidate target for melanoma treatment, no selective inhibitor of this kinase is currently available. Among the other resistance-associated genes identified in this study, *BCL2A1*/

---

lines. **(H)** Spare respiratory capacity in human melanoma cell lines. **(I)** Maximal induction of ECAR by oligomycin and monensin relative to basal conditions in melanoma cells. **(J)** Glycolytic reserve in human melanoma cell lines. **(K)** Characterization of Rho-0 MeWo cells by comparison of mitochondrial features with parental MeWo cells. **(L)** Cell viability assessed with PrestoBlue HS, expressed as % of maximal viability. Cells were treated for 72 h with 10-fold dilutions of BI 6727. **(M)** Viability of cells treated with 10 µM BI 6727 for 24 h. **(N, O)** *CXCL8* and *IL6* transcript levels in MeWo and MeWo-Rho-0 cells treated DMSO 0.1% and inhibitors at the indicated doses for 72 h. Data are shown as mean ± SEM of three or more independent experiments. **(D, E, F, G, H, I, J)**: *$P < 0.05$, one-way ANOVA with Tukey's multiple comparison test; **(K, M, N, O)**: *$P < 0.05$, paired *t* test, except for mtDNA/gDNA data in (K) (unpaired *t* test).

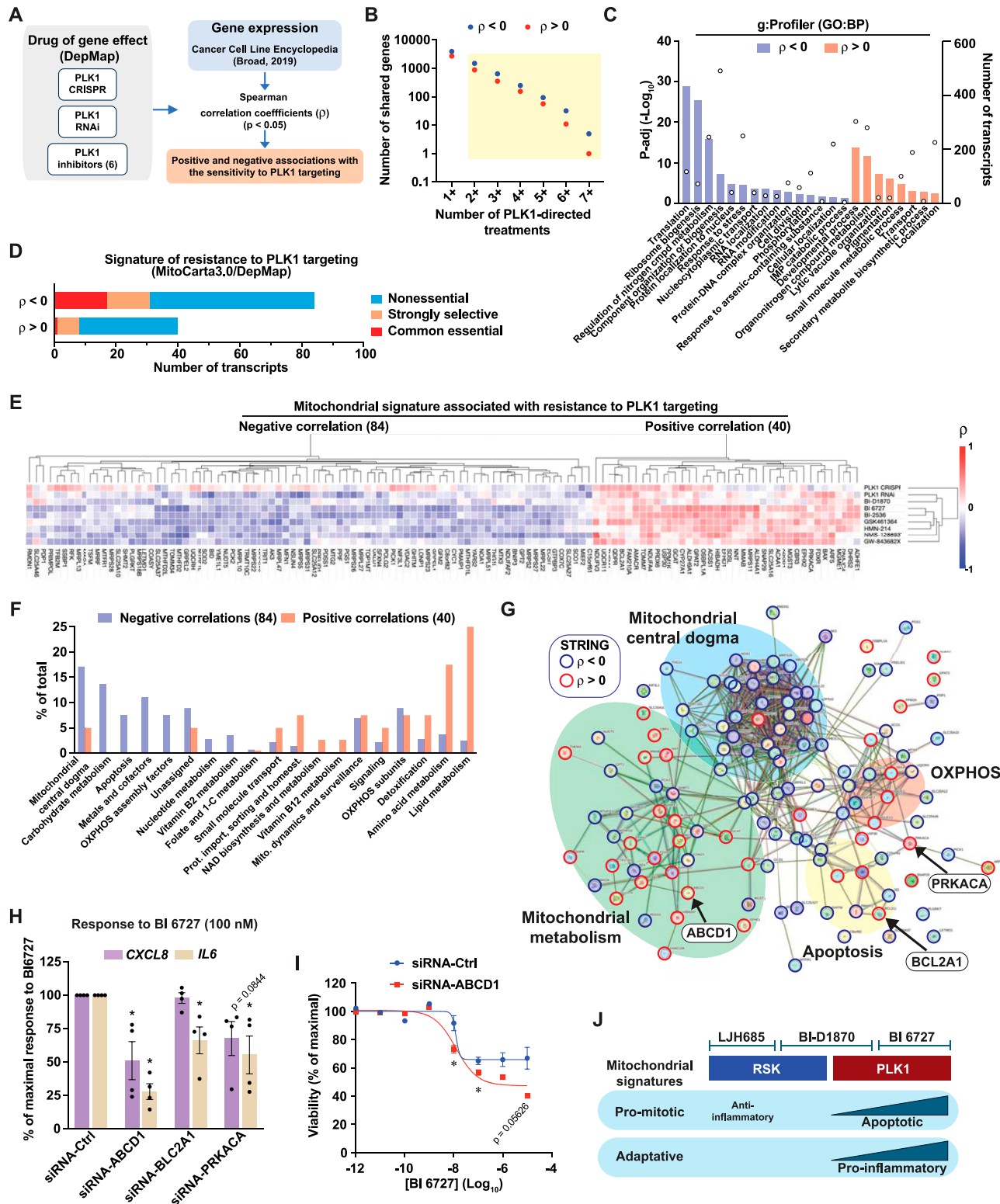

**Figure 7. Defining actionable mitochondrial signatures associated with resistance to PLK1 targeting in melanoma.**

See also Fig S5. **(A)** Schematic depicting the strategy to identify transcripts associated with resistance to PLK1 targeting using the indicated publicly available datasets. **(B)** Transcripts that significantly correlate (*P* < 0.05) with two or more PLK1-directed treatments were selected for further analyses. The following publicly available datasets were used: Cancer Cell Line Encyclopedia (Broad, 2019); DepMap Portal: PLK1 CRISPR (DepMap Public 24Q2+Score, Chronos), PLK1 RNAi (Achilles+DRIVE+Marcotte, DEMETER2), Drug sensitivity data for the PLK1 inhibitors BI 6727, BI-2536, GSK-461364, HMN-214, NMS-1286937, GW-843682X (PRISM repurposing primary screen). **(C)** Functional gene analysis (g:Profiler, GO:BP) performed on transcripts that correlate with the sensitivity to PLK1 targeting. Bars indicate p-adj values and circles indicate

Bfl-1 encodes a Bcl-2-related antiapoptotic protein that interacts with multiple proapoptotic proteins and blocks the release of cytochrome c (reviewed in Vogler [2012]). In melanoma, it has been shown that this gene is regulated transcriptionally by MITF and that its expression is associated with resistance to BRAF inhibition (Haq et al, 2013b). Although no highly selective BCL2A1 inhibitor has yet been identified (Li et al, 2021), a few inhibitors have shown promising results in experimental cancer studies (D de Araujo et al, 2018; Haq et al, 2013b), including the BET inhibitor CPI-0610, which indirectly targets the transcription of BCL2A1 (Yamatani et al, 2022). Our analyses also revealed that the ABC transporter ABCD1 is linked to resistance to PLK1 targeting in melanoma. ABCD1 is critical for the peroxisomal import of very long fatty acids and related CoA esters for catabolism through β-oxidation, and mutations in this gene are responsible for adrenoleukodystrophy, an X-linked and neurode-generative disorder associated with microglial apoptosis (Eichler et al, 2008). In astrocytes and oligodendrocytes, silencing of ABCD1 impaired mitochondrial respiration and induced mitochondrial superoxide anion production (Baarine et al, 2015), suggesting that this gene could exert protective and antiapoptotic functions in melanoma cells. In addition to these precision oncology ap-proaches, our study also shows that the depletion of mtDNA in MeWo cells dramatically alters their resistance profile to PLK1 in-hibition. Rho-0 cells still possess mitochondria, but they are part of fragmented networks and show no respiratory activity. As our data indicate that OXPHOS is not associated with resistance, it would be interesting to determine which of the genes from our PLK1-targeting resistance signature are expressed in Rho-0 cells and whether this model can be used to generate resistant cells. Interestingly, Rho-0 cells were more resistant to low doses of PLK1 inhibitor than parental cells. We believe that this resistance may be linked to the slower division rate of Rho-0 cells, which is in line with the fact that PLK1 inhibition is more potent in rapidly dividing cells. It is important to specify that our Rho-0 model has several limitations including the continuous use of ethidium bromide (EtBr) to maintain low levels of mtDNA, which can lead to unwanted mutations in genomic DNA and effects unrelated to mitochondria. However, a concentration of 100 ng/ml is among the lowest concentrations used in the literature for the generation of Rho-0 cells (50–500 ng/ml). The impact on viability was low, with Rho-0 cells having ~14% lower viability than parental MeWo cells, and it is not possible to segregate genotoxic and mitochondrial-specific effects. The knock-out/knockdown of mitochondrial biogenesis factors such as TFAM could represent alternative approaches in future studies (Larsson et al, 1998).

In this study, we also show that the antiproliferative and pro-inflammatory effects of unselective RSK inhibitors are likely due to the inhibition of other kinases, such as PLK1. Strikingly, BI-D1870 and BI 6727 have similar pro-inflammatory effects in MeWo cells, but they are, respectively, anti-inflammatory and proapoptotic in A-375 cells (Figs 1D and 4A). This suggests that the anti-inflammatory effects of BI-D1870 in A-375 cells (Fig 1D) are linked to RSK inhibition, as observed with selective RSK inhibitors such as LJH685 and LJI308 (Fig 1D and H). Indeed, despite its impact on PLK1 or other kinases, treatment with BI-D1870 also shares with LJH685 the induction of antigen presentation genes (Figs S2A and 3G–K). One important observation of our study is that specific inhibition of RSK does not typically induce CXCL8 and has a mild impact on proliferation. We have previously shown that RSK inhibitors can reduce glycolytic flux by directly phosphorylating PFK-2 (Houles et al, 2018). Since RSK is involved in the control of mRNA translation and is also a potent modulator of inflammation through the TRAF6-IKK-NF-κB pathway (Zhang et al, 2005; Peng et al, 2010; Yao et al, 2018), we hypothesize that RSK inhibition in melanoma may globally reduce the bioenergetic charge and associated oxidative stress in melanoma cells. Interestingly, RSK targeting by RNA interference was shown to suppress proliferation both in vitro and in vivo (Theodosakis et al, 2017; Houles et al, 2018). While the context-dependent roles of RSK may explain these discrepancies, kinase-independent functions of RSK through inhibitory interactions with CBP and ERK may explain the differences between RSK inhibition and gene/RNA silencing (Merienne et al, 2001; Kim et al, 2006).

Phenotype switching is the transition of melanoma cells from differentiated/melanocytic/proliferative states to dedifferentiated, poorly proliferative, migratory and invasive states in response to hypoxia, pro-inflammatory factors, and therapies (O'Connell et al, 2013; Saez-Ayala et al, 2013; Widmer et al, 2013; Kemper et al, 2014; Riesenberg et al, 2015; Rambow et al, 2018). Here, we show that pharmacological targeting or KD of PLK1 in MeWo cells represses the expression of melanocytic transcripts while increasing the ones of invasive and pro-inflammatory transcripts, supporting a switch toward dedifferentiation. A-375 cells do not acquire a dediffer-entiated phenotype and instead show greater sensitivity to PLK1 targeting. Our results suggest that A-375 cells are poorly equipped to adapt to PLK1 inhibition and thus engage apoptosis. Indeed, targeting PLK1 by RNA interference or small inhibitors in A-375 cells has been shown to repress the expression of multiple metabolic proteins involved in glycolysis and the TCA cycle (Gutteridge et al, 2017). Compared with MeWo cells, A-375 cells were shown to express lower levels of the mitochondrial biogenesis coactivator PGC-1α and several antioxidant enzymes (Vazquez et al, 2013; Torrens-Mas et al, 2017). This high sensitivity of A-375 cells to PLK1 inhibiton may be linked to their BRAF V600E status since this form of mutated BRAF inhibits the MITF/PGC-1α axis and lowers the cellular ability to

---

number of transcripts associated with a gene set. **(D)** Essentiality of transcripts encoding for mitochondrial proteins that significantly correlate with resistance to PLK targeting in melanoma cells (DepMap). **(E)** Heat-map representation and hierarchical clustering of Spearman correlations between the 124 mitochondrial signature transcripts and respective impact on proliferation dynamics of BI-D1870 and PLK1-targeting approaches (DepMap). **(F)** Functional classification of 124 mitochondrial signature transcripts associated with resistance to PLK1 targeting, according to MitoPathways3.0. **(G)** STRING interaction network of the 124 mitochondrial signature transcripts associated with resistance to PLK1 targeting. Nodes indicate proteins, and edges indicate interactions. Spearman's correlation with proliferation dynamics (DepMap) is indicated by nodes' colored borders. **(H)** Gene expression in MeWo cells transfected with the indicated siRNA for 48 h and treated with DMSO 0.1% or BI 6726 (100 nM) for 72 h. **(I)** Viability of MeWo cells assessed with PrestoBlue HS, expressed as % of maximal viability. MeWo cells were treated with 10-fold dilutions of BI 6727. **(J)** Model depicting the impact of mitochondrial signatures on the response to RSK and PLK inhibitors. Data are shown as mean ± SEM of three independent experiments. *$P <$ 0.05, one-way ANOVA with Dunnett multiple comparison test.

resist oxidative stress (Haq et al, 2013a). While our study reveals a marked heterogeneity in the pro-inflammatory response to RSK/PLK1 inhibitors in nine human melanoma cell lines (Fig 1D), most of our experiments compare only two cell lines, MeWo and A-375. Unlike A-375 cells, MeWo cells are BRAF wild-type and are characterized by NF1 loss (hemizygous Q1336* nonsense mutation) (Nissan et al, 2014). NF1 mutations are frequent in melanoma and have been associated with resistance to directed therapies (Maertens et al, 2013; Nissan et al, 2014; Kiuru & Busam, 2017). Also, opposite to A-375, MeWo cells are classified as MITF and PGC-1α positive cells (Vazquez et al, 2013), which aligns well with the importance of mitochondrial programs in their resistance phenotype. While MITF has been shown to antagonize pro-inflammatory responses through inhibition of NF-κB and AP-1 in melanoma (Riesenberg et al, 2015), we did not observe a reduction in MITF and PGC-1α transcript in MeWo cells treated with the RSK/PLK1 inhibitor BI-D1870. We have identified two additional cell lines that are pro-inflammatory in response to RSK/PLK1 inhibition: IGR-1 (hemizygous BRAF V600K) and SK-MEL-5 (hemizygous BRAF V600E) (Fig 1D). This suggests that the sensitivity of A-375 cells to RSK/PLK inhibitors may not be fully explained by their mutated BRAF status. Our analyses also indicate that basal profiling of mitochondria (mtDNA content, expression of OXPHOS proteins, respiration) does not correlate with the sensitivity to RSK/PLK1 inhibition in six melanoma cell lines. Strikingly, A-375 cells responded to the uncoupler FCCP and the ionophore monensin by a dramatic shift towards glycolysis compared with the five other tested cell lines, which may indicate a metabolic vulnerability that explains their sensitivity to PLK1 inhibition. On the other hand, the BRAF V600 mutated COLO 829 and WM983B cells, in which RSK/PLK1 inhibition had anti-inflammatory effects as in A-375, did not respond markedly to FCCP and monensin. In future studies, it will be necessary to determine if resistance mechanisms are linked to an ability to maintain mitochondrial functions under PLK1 inhibition. It is likely that a wide variety of mechanisms may explain the involvement of mitochondrial proteins in resistance to PLK1 targeting. A chemogenomic screen coupling PLK1 inhibition and genome-wide gene knockout in melanoma cells could be useful to precisely identify sensitization or desensitization factors that could explain therapeutic resistance or help better position PLK1 inhibitors.

In conclusion, our study shows that the expression levels of mitochondrial proteins within a signature of hundreds of genes define resistance or sensitivity to therapeutic approaches targeting PLK1. It remains to be determined whether genes within this signature predict response to PLK1 inhibitors or the development of resistance, and to characterize the mechanisms by which these proteins confer resistance. Although our study shows that cells resistant to PLK1 targeting adopt a pro-inflammatory dedifferentiated phenotype, it will be needed to study this response in a larger number of cell lines. In addition, it will be important to document other aspects of this phenotype such as invasive and metastatic properties, and the impact on the efficacy of other targeted therapies and immunotherapy. Furthermore, our analysis of selective RSK inhibitors suggests that the antitumor effects of nonselective RSK inhibitors are due to PLK1 inhibition. Ultimately, our research paves the way for the identification of mitochondrial therapeutic targets and their rational use in combination with targeted therapies to increase therapeutic efficacy and minimize the development of resistance.

# Materials and Methods

### Cell lines and treatments

Human melanoma cell lines A-375, MeWo, SK-MEL-5, IGR-1, RPMI-7951, COLO 829, Malme-3M, WM983B, and Mel JuSo were kindly provided by Pr. Ian Watson (McGill University). All cell lines, including MeWo-Rho-0, were authenticated by comparing STR profiles with the ATCC and Cellosaurus databases. All cell lines tested negative for mycoplasma using a PCR mycoplasma detection kit (#G238; Applied Biological Materials Inc.). Cell lines were cultured in RPMI-1640 medium (#350-000-CL; Wisent Inc.) supplemented with 10% FBS (#090-150; Wisent Inc.) and 1% Penicillin/Streptomycin (#450-201-EL; Wisent Inc.). Cells were maintained at 37°C in a humidified incubator with 5% $CO_2$. Rho-0 cells were generated from parental MeWo cells cultured in media containing 50 μg/ml uridine (#URD222; Bioshop Canada Inc.), 100 μg/ml sodium pyruvate (#600-110-EL; Wisent Inc.), and 100 ng/ml ethidium bromide (#ETB333; Bioshop Canada Inc.) for several weeks. The rho-0 medium was replaced with regular media a few days before the experiments. Rho-0 cells grew slightly slower than parental MeWo cells and showed a ~14% decrease in viability at baseline, as assessed by trypan blue exclusion assay.

Inhibitors and compounds were resuspended in DMSO (#BP231-100; Thermo Fisher Scientific), as recommended by the manufacturer, and stored in a −80°C freezer. Before experiments, compound stocks were further diluted in DMSO to obtain a final DMSO concentration of 0.1% (vol/vol) in culture media. BI-D1870 (#15264), BI 6727 (#18193), BIX 02565 (#19183), BRD7389 (#20214), Cdk2 inhibitor II (#15154), DCC-2036 (#21465), GSK2256098 (#22995), LJH685 (#19913), LJI308 (#19924), MRT68921 (#19905), PF-03814735 (#15015), SU11274 (#14861), and WHI-P154 (#16178) were purchased from Cayman Chemical. Doxorubicin hydrochloride (ab120629) was purchased from Abcam Inc.. Recombinant human IFN-γ (#300-02) was from PeproTech and stored at 100 μg/ml (10% (vol/vol) FBS in sterile water) in a −80°C freezer.

### Gene silencing

A-375 and MeWo cells were seeded in six-well culture plates with 2 ml of media per well at cell densities of 15,000 cells/ml and 75,000 cells/ml, respectively. Cells were incubated for 16–24 h before transfection with siRNA. Non-targeting siRNA (1027280; Allstar) and siRNA pools, made of four siRNAs (1027416; FlexiTube Gene Solutions) against human PLK1 (GS5347), ABCD1 (GS215), BCL2A1 (GS597), and PRKACA (GS5566) were purchased from QIAGEN. Cells were transfected with Lipofectamine RNAiMAX transfection reagent (#13778500; Thermo Fisher Scientific) using Opti-MEM low serum medium (#31985070; Thermo Fisher Scientific). The final concentration of the non-targeting control siRNA and siRNA pools was 10 nM (2.5 nM for each siRNA used for the pools), except for PLK1, which was 1 nM. Transfections were performed according to the

manufacturer's protocol, using 5 $\mu$l of transfection reagent per well. The medium was replaced 24 h post-transfection.

## Cell counting and viability assay

Cells were seeded in six-well culture plates in 2 ml of media at the following densities: 9,375 cells/ml (A-375), 50,000 cells/ml (Mel JuSo, SK-MEL-5), 75,000 cells/ml (MeWo, Malme-3M), 70,000 cells/ml (RPMI-7951), 62,500 cells/ml (COLO 829), 30,000 cells/ml (IGR-1), and 25,000 cells/ml (WM983B). Cells were incubated for 16–24 h before treatments. 72 h posttreatment, cells were rinsed with PBS 1X (#311-010-CL; Wisent Inc.), dissociated with Trypsin/EDTA (0.25% Trypsin and 2.21 mM EDTA 4Na, #325-043-EL; Wisent Inc.), and resuspended in 1 ml of RPMI-1640 medium (#350-000-CL; Wisent Inc.). Cell suspensions were mixed with trypan blue solution (0.4% solution in phosphate buffer saline, #609-130-EL; Wisent Inc.) and counted manually using a hemacytometer (#1492; Hausser Scientific). For proliferation curves, cell culture media were changed every day with the renewal of treatments. Occasionally, cell suspensions were centrifuged at 1,000$g$ for 5 min and resuspended in a smaller volume of culture medium.

For the assessment of viability, cells were seeded in 96-well plates at 1,250 cells/100 $\mu$l (A-375), 7,500 cells/100 $\mu$l (MeWo), 10,000 cells/100 $\mu$l (Rho-0) to which 100 $\mu$l of 2X concentrated inhibitor dilutions (BI-D1870, LJH685, and BI 6727) were added. Six wells per plate contained only the cell culture medium (no cell control). Cell viability was measured 72 h later with the PrestoBlue HS cell viability reagent (#P50201; Thermo Fisher Scientific) following the manufacturer's protocol. The absorbance was measured using a Varioskan LUX multimode microplate reader (Thermo Fisher Scientific). Viability is calculated as percentage of maximal viability since PrestoBlue HS cell viability signal is proportional to viable cell count only. Thus, maximal viability is set at 100% using the average absorbance values of the lowest doses of BI 6727 (0.001, 0.01, 0.1, 1.0 nM), which were indistinguishable from DMSO 0.1% values (cannot be plotted on a logarithmic scale) in all independent experiments. All other doses are relative to this average value and thus are represented as %. For crystal violet staining, cells were seeded in 24-well culture plates in 500 $\mu$l of media at the following densities: 7,500 cells/ml (A-375) and 45,000 cells/ml (MeWo). Cells were incubated for 16–24 h before treatments. 72 h posttreatment, cells were rinsed with PBS 1X and fixed with 10% formalin (vol/vol in water) for 15 min. Cells were rinsed with water and stained with 0.5% crystal violet solution (wt/vol in 20% methanol) for 30 min. Cells were rinsed with water, and the plates were allowed to dry completely. Images were taken with a ChemiDoc XRS+ using Image Lab software (Bio-Rad). Dye extraction was performed with 33% (vol/vol) acetic acid solution for 30 min and absorbance at 590 nm was measured using a Varioskan LUX multimode microplate reader (Thermo Fisher Scientific).

## Gene expression analysis

Total cellular RNA was isolated using the Monarch Total RNA Miniprep Kit (#T2010; NEB) following the manufacturer's protocol. The RNA concentration was determined using a Varioskan LUX multimode microplate reader (Thermo Fisher Scientific) with a $\mu$Drop plate (Thermo Fisher Scientific). Reverse transcription was

performed using the LunaScript RT SuperMix Kit (#E3010; NEB) according to the manufacturer's protocol on a SimpliAmp thermal cycler (Applied Biosystems). Oligonucleotide primers were designed using Primer Blast software (NCBI), synthesized by Integrated DNA Technologies (IDT), and validated for efficiency and specificity. The list of primers for the detection of human transcripts by qRT-PCR is as follows (5'→3' direction):
 *TBP* forward: TGCCACGCCAGCTTCGGAGA
 *TBP* reverse: ACCGCAGCAAACCGCTTGGG
 *UBE2D2* forward: AGAGAATCCACAAGGAATTGAATGA
 *UBE2D2* reverse: TAGGGACTGTCATTTGGCCC
 *CXCL8* forward: TGATTTCTGCAGCTCTGTGT
 *CXCL8* reverse: AAACTTCTCCACAACCCTCT
 *IL6* forward: CAGAGCTGTGCAGATGAGTA
 *IL6* reverse: GCGCAGAATGAGATGAGTTG
 *HLA-DOA* forward: GGGGTTCCACACCCTGATGA
 *HLA-DOA* reverse: CGCCGTAAGACTGGTAGAAGG
 *HLA-DQA2* forward: CCCTGTGGAGGTGAAGACAT
 *HLA-DQA2* reverse: AGTCTCTTTCGTCTCCAGGTC
 *CD74* forward: TTGGAGCAAAAGCCCACTGA
 *CD74* reverse: GCACTTGGGCCTGAATGAAC
 *ERAP1* forward: TTTCACTTTCGGTCCTGGGG
 *ERAP1* reverse: TGAGGGGCAGAAACACCATC
 *PLK1* forward: AGTACGGCCTTGGGTATCAG
 *PLK1* reverse: GCTCGCTCATGTAATTGCGG
 *BCL2A1* forward: ACCAGGCAGAAGATGACAGA
 *BLC2A1* reverse: TGGTATCTGTAGGACGCACT
 *ABCD1* forward: ACTCAGTGGAGGACATGCAA
 *ABCD1* reverse: ACGTCCTTCCAGTCACACAT
 *PRKACA* forward: CGAGCAGGAGAGCGTGAAAGA
 *PRKACA* reverse: CCAAGTGGGCTGTGTTCTGA
 *TYRP1* forward: CATGCAGGAAATGTTGCAAGAA
 *TYRP1* reverse: AGTTTGGGCTTATTAGAGTGGAATC
 *DCT* forward: GTCTGTGGCTCTCAGCAAGG
 *DCT* reverse: ATAGCCGGCAAAGTTTCCTGT
 *TYR* forward: AGCTATCTACAAGATTCAGACCCAG
 *TYR* reverse: TGACGACACAGCAAGCTCA
 *PMEL* forward: TCCAGGCTTTGGTTCTGAGT
 *PMEL* reverse: TGTGATAGGTGCTTTGCTGG
 *MLANA* forward: GCTCATCGGCTGTTGGTATT
 *MLANA* reverse: AGAGACACTTTGCTGTCCCG
 *MYC* forward: TACTGCGACGAGGAGGAGAA
 *MYC* reverse: CGAAGGGAGAAGGGTGTGAC
 *IRF4* forward: CCCGGAAATCCCGTACCAAT
 *IRF4* reverse: AGGTGGGGCACAAGCATAAA
 *KIT* forward: AACACGCACCTGCTGAAATG
 *KIT* reverse: GTCTACCACGGGCTTCTGTC
 *TGFB1* forward: CGCGTGCTAATGGTGGAAAC
 *TGFB1* reverse: GCTGAGGTATCGCAAGGAAT
 *DKK1* forward: TCACGCTATGTGCTGCCC
 *DKK1* reverse: TGACCGGAGACAAACAGAACC
 *NGFR* forward: AGGGAGGAATCGAGGAACCA
 *NGFR* reverse: TCTGGCTTTGGGCGAATCAT
 *PTX3* forward: AGTGCCTGCATTTGGGTCAA
 *PTX3* reverse: CTCTCCACCCACCACAAACA
Quantitative RT–PCR was performed with the Luna Universal qPCR Master Mix (#M3003; NEB) using the QuantStudio 5 Real-time

PCR system (Applied Biosystems). The PCR program was as follows: 95°C for 3 min followed by 45 cycles at 95°C for 15 s and 60°C for 30 s. Melting curve analysis was 95°C for 15 s, 60°C for 1 min, and an increase in temperature to 95°C at 0.1°C/s. QuantStudio Design and Analysis (Applied Biosystems) was used for data analysis. Gene expression levels were calculated using the ΔΔCt method, in which RNA levels were normalized against *TBP* or *UBE2D2*.

## Quantification of mitochondrial DNA

Cells were lysed in SNET lysis buffer (20 nM Tris–HCl pH 8.0, 5 mM EDTA pH 8.0, 400 mM NaCl, 1% (wt/vol) SDS, 100 µg/ml RNAse A, and 400 µg/ml proteinase K) for 16–20 h at 55°C on a digital block heater. Cellular DNA was extracted with one volume of phenol: chloroform:isoamyl alcohol (25:24:1, #15593031; Thermo Fisher Scientific), precipitated with one volume of isopropyl alcohol, and centrifuged at 17,000$g$ for 10 min. Pellets were washed with 70% ethanol and allowed to dry at RT before being dissolved in TE buffer (1 mM EDTA pH 8.0, 10 mM Tris–HCl pH 8.0) for 16–20 h at 4°C. The same procedure was used for sample preparation for STR profiling. DNA concentrations were measured using a Varioskan LUX multimode microplate reader (Thermo Fisher Scientific). Mitochondrial and genomic DNA were quantified using quantitative PCR. The PCR program was: 95°C for 6 min, followed by 45 cycles of 95°C for 15 s and 60°C for 30 s. The following list of primers was used to amplify human mitochondrial and genomic DNA (5′→3′ direction):

*MT-CYB* (mitochondrial DNA) forward: GCGTCCTTGCCCTATTACTATC
*MT-CYB* (mitochondrial DNA) reverse: CTTACTGGTTGTCCTCCGATTC
*RPL13A* (genomic DNA) forward: CTCAAGGTCGTGCGTCTG
*RPL13A* (genomic DNA) reverse: TGGCTTTCTCTTTCCTCTTCTC

## Western blotting

Cells were washed with ice-cold PBS and lysed with RIPA low SDS buffer (50 mM Tris–HCl pH 7.4, 150 mM NaCl, 5 mM EDTA pH 8.0, 1 mM EGTA pH 8.0, 1% NP-40, 0.5% sodium deoxycholate, 0.1% SDS) supplemented with sodium fluoride, sodium orthovanadate, and protease inhibitor cocktail (ab270055; Abcam Inc.). Lysates were frozen and thawed three times in liquid nitrogen to maximize lysis efficiency and kept on ice for 30 min. Samples were centrifuged at 17,000$g$ for 10 min at 4°C, and protein concentration was determined with the Pierce BCA Protein Assay Kit (#23225; Thermo Fisher Scientific). Equal amounts of total proteins (10 µg) were denatured in sample buffer (50 mM Tris–HCl pH 6.8, 100 mM DTT, 2% SDS, <1.5 mM bromophenol blue 1.5 mM, 1.075 M glycerol) at 95°C for 5 min. Denatured proteins were resolved on 8–12 polyacrylamide-SDS gels and transferred onto PVDF membranes (GE Healthcare). A molecular weight marker was added to each gel for reference (Precision Plus Protein Dual Color, #1610374; Bio-Rad). Transfer on PVDF membranes was performed using Pierce Western blot transfer buffer (#35040; Thermo Fisher Scientific) and Trans-Blot SD semidry transfer cell (Bio-Rad) at 22 V and 0.2 A for 50 min. Blots were blocked with 5% non-fat dry milk diluted in TBS supplemented with 0.1% Tween-20 (TBST) and incubated with primary antibodies overnight at 4°C with gentle rocking. Primary antibodies against PARP (#9542), β-actin (#3700), PLK1 (#4535), and Melan-A/MART-1/MLANA (#64718) were purchased from Cell Signaling

Technology Inc.. Antibodies against the OXPHOS protein complexes (#45-8099) and Vinculin (#MA5-11690) were purchased from Thermo Fisher Scientific. Antibodies against dopachrome tautomerase/DCT (#HPA010743) and tyrosinase/TYR (#05-647) were purchased from MilliporeSigma. The membranes were washed five times for 5 min with TBST and incubated with anti-mouse or anti-rabbit secondary antibodies coupled with horseradish peroxidase at 1:10,000 dilution (#KP-5220-0341, #KP-5220-0336, Mandel Scientific) at RT for 1 h. Samples were revealed using the SuperSignal West Pico PLUS Chemiluminescent substrate (#34579; Thermo Fisher Scientific). Chemiluminescence was detected on Blu-Lite films (Dutscher). The films were developed in a dark room by using an X-ray film processor. Densitometric analyses were performed using ImageJ software (NIH).

## Cytokine array

MeWo cells were seeded in six-well plates (2 ml/well) at density of 75,000 cells/ml. Cells were incubated for 16–24 h before treatment. 48 h posttreatment, the medium was replaced with fresh medium containing the same concentration of the corresponding inhibitor. Conditioned media from the treated cells were collected 72 h posttreatment and cleared by centrifugation at 1,000$g$ for 5 min at 4°C. Cytokine concentrations were quantified using the Human Cytokine Antibody Array following the manufacturer's protocol (#ab133997; Abcam Inc.). 1 ml of clarified media was applied per cytokine membrane, followed by overnight incubation at 4°C, and chemiluminescence was detected on Blu-Lite films (Dutscher) and developed in a dark room using an X-ray film processor. Densitometry was performed using the Protein Array Analyzer extension of the ImageJ software (NIH). The spot densitometric values were normalized and corrected for positive and negative controls. Densitometric values were corrected for cell counts.

## Oxygen consumption and extracellular acidification

Oxygen consumption rates (OCRs) and extracellular acidification rates (ECARs) were measured using an XFe96 Seahorse extracellular analyzer (Agilent Technologies Inc.). Seahorse XFe96 cell cultures microplates (Agilent Technologies Inc.) were pre-coated with 50 µg/ml Poly-D-Lysine (#A3890401; Thermo Fisher Scientific) in D-PBS (#311-425-CL; Wisent inc.). The Seahorse XFe96 sensor cartridges (Agilent Technologies Inc.) were calibrated according to the manufacturer's protocol. Assay media were prepared as follows, for 100 ml: 1 g RPM1640 (with glucose, without L-glutamine and sodium bicarbonate, #260-012 XK; Wisent inc.), 29.2 mg L-glutamine (#AC386032500; Fisher Chemical), 100 µl HEPES 1 M (#330-050; Wisent Inc.), 100 ml water, with pH at 7.2. Cells were plated in poly-D-lysine-coated microplates at a density of 45,000 cells/well (A-375), 60,000 cells/well (SK-MEL-5), 70,000 cells/well (MeWo), 60,000 cells/well (WM983B), 65,000 cells/well (IGR-1), 65,000 cells/well (COLO 829), and 70,000 cells/well (MeWo-Rho-0) in 100 µl of assay medium. The default values for the measurement steps were selected using Wave software (Agilent Technologies Inc.). Sequential measurement steps were: (1) baseline (no injection), (2) oligomycin 3 µM (#ab141829; Abcam Inc.), (3) FCCP 0.5 µM (#15218;

Cayman Chemical), (4) rotenone 1 $\mu$M (#ab143145; Abcam Inc.) and antimycin A 1 $\mu$M (#ab141904; Abcam Inc.), and (5) monensin 20 $\mu$M (#M5273; MilliporeSigma). OCR and ECAR were normalized with cell counts. Non-mitochondrial respiration was defined as respiration in cells treated with rotenone and antimycin A, and was subtracted from all OCR values. Uncoupled respiration was determined after oligomycin injection. Maximal respiration was determined after FCCP injection. Spare respiratory capacity was defined as the maximal respiration minus basal respiration (untreated). The maximal ECAR was determined after monensin treatment. Glycolytic reserve was defined as the maximal ECAR minus the basal ECAR (untreated). For the MeWo and MeWo-Rho-0 comparisons, basal (untreated) OCR and ECAR in MeWo were set at 100%.

### RNA-seq

RNA was quantified using a Qubit (Thermo Fisher Scientific), and quality was assessed using a 2100 Bioanalyzer (Agilent Technologies). Transcriptome libraries were generated using KAPA RNA HyperPrep (Roche) using poly-A selection (Thermo Fisher Scientific). Sequencing was performed on an Illumina Next-Seq500 (Illumina), obtaining ~30 M single-end 75 bp reads per sample. The samples from the MeWo/BI-D1870 experiments were run independently of those from the A-375/LJH685 experiments. Sequences were trimmed for sequencing adapters and low-quality 3' bases using Trimmomatic version 0.35 (Bolger et al, 2014) and aligned to the reference human genome version GRCh38 (gene annotation from Gencode version 37, based on Ensembl 103) using STAR version 2.7.1a (Dobin et al, 2013). Gene expression was obtained as read counts directly from STAR and computed using RSEM (Li & Dewey, 2011) to obtain normalized gene and transcript level expression, in TPM values, for these stranded RNA libraries. DESeq2 version 1.30.1 (Love et al, 2014) was used to normalize gene read counts. Sample clustering and principal component analysis (PCA) showed clear and unambiguous clustering of the DMSO and RSK inhibitor (BI-D1870, LJH685)-treated groups. Using DESeq2, we introduced a second factor into the model to consider the sample pairing effect by using a likelihood-ratio test to compare the full model and a reduced model with the goal of isolating the condition effect. After removing the pairing effect contribution, we obtain a PCA showing less variance in the PC2 component, representing variability between replicates. Thus, all RNA-seq analyses considered the batch effect of independent experiments.

### Functional classification and gene set enrichment analyses

Threshold values for fold change > 1.2-fold and false-discovery rate (FDR)-adjusted $P$-values < 0.1 were used to select differentially expressed transcripts (DETs) between groups treated with RSK inhibitors (BI-D1870, LJH685) versus control (DMSO). $Log_2$ fold changes and adjusted $P$ values were calculated using DESeq2. The g:Profiler web-based software (Kolberg et al, 2023) was used for the functional classification of DET based on gene sets from the Reactome pathway database (Milacic et al, 2024). Gene set enrichment analyses (GSEA) were performed using the GSEA software (Mootha et al, 2003; Subramanian et al, 2005) (v.

4.3.0) using normalized read counts (DESeq2). Undetected genes (0 reads) in all samples were not included in the analyses. The Hallmarks MSigDB gene set collection was used for GSEA. Visual representation of GSEA gene set networks was performed using Cytoscape software (Shannon et al, 2003) (v. 3.10.1) with the EnrichmentMap application (Merico et al, 2010). The Molecular Signature DataBase (MSigDB) was used for GSEA and network visualization. Enrichment map cutoff parameters were $P$ value = 0.05 and FDR Q value = 0.1. Isolated groups of nodes and edges could have been relocated without modification to increase the comprehensibility of the networks. Colored regions were manually generated to highlight specific groups of linked gene sets. The figures do not include small groups (e.g., 4 nodes or less).

### Analysis of publicly available databases

RNA expression datasets in human tumors and cell lines were obtained from the open-access and open-source resource cBio-Portal (Cerami et al, 2012; Gao et al, 2013). The human cutaneous melanoma transcriptomics dataset (SKCM) shown here was obtained from TCGA Research Network: http://www.cancer.gov/tcga (PanCancer Atlas). The human melanoma cell line transcriptomic dataset was obtained from the Cancer Cell Line Encyclopedia (CCLE, Broad, 2019) (Ghandi et al, 2019). Perturbation effects associated with CRISPR-Cas9 (Public 23Q4+Score, Gene effect: Chronos), RNAi (Achilles+DRIVE+Marcotte, Gene effect: DEMETER2), and inhibitors (PRISM Repurposing Primary Screen, Drug sensitivity/effect: $Log_2FC$ of cell count) in human melanoma cell lines were obtained from the DepMap Portal: https://depmap.org/portal. Spearman correlation coefficients ($\rho$) between genes of the SKCM dataset were obtained from cBioPortal. Spearman correlation coefficients between perturbation effects (DepMap) and transcript levels (CCLE) were calculated using Prism (GraphPad Software). Gene essentiality was determined using DepMap. The MitoCarta3.0 (Rath et al, 2021) dataset and associated MitoPathways3.0 were used to identify and classify mitochondrial proteins associated with transcriptomic data. Heat-maps were generated using Morpheus: https://software.broadinstitute.org/morpheus. For hierarchical clustering and dendrograms, the metric used was Euclidean distance, the linkage method was average, and clustering was performed on rows and columns.

### Interaction networks

STRING web-based resource (Szklarczyk et al, 2015) was used to generate an interaction network using the 124-transcript signature associated with resistance to PLK1-targeting approaches (CRISPR-Cas9, RNAi, inhibitors). The mapping settings were set to default: full STRING network (the edges indicate functional and physical associations), meaning of edges based on evidence, with a minimum confidence score of 0.400 (111 proteins involved in interactions, 13 non-interacting proteins). Colored regions were manually generated to highlight enriched networks. The network statistics were as follows: 541 edges, average node degree of 8.73, average local clustering coefficient of 0.497, and PPI enrichment $P$ value of < 1.0 × $10^{-16}$.

## Statistical analyses

Statistical analyses were performed using the Prism software 10_v. 10.2.3 (GraphPad) or the Microsoft Excel software. All data shown represent the combination mean values obtained from at least three independent experiments performed at different times with different cell passages. Paired statistical analyses refer to the pairing of samples within independent experiments. Unpaired analyses refer to independent experiments that cannot be paired, such as different cell lines that have not been tested simultaneously.

## Data Availability

The raw and processed data from RNA-seq analyses in this study have been deposited to the NCBI GEO database (https://www.ncbi.nlm.nih.gov/geo/) and assigned the identifier GSE264686. Additional supporting data are available from the corresponding author upon reasonable request.

## Supplementary Information

## Acknowledgements

The authors thank the IRIC genomic platform and IRIC bioinformaticians for assistance with RNA-seq. The authors thank the Centre for Applied Genomics (The Hospital for Sick Children, Toronto) for STR profiling. The authors thank Caroline Thivierge, Sarah-Slim Diwan, Hamza Haddouch, and Rita Maria Kenaan El Rahbani for editing the manuscript. This work was supported by the Cancer Research Society (Grant 840633, 2021, Grant 1054571, 2023) and the National Sciences and Engineering Research Council of Canada (Grant RGPIN-2023-03828, 2023).

### Author Contributions

É Lavallée: validation, investigation, methodology, and writing—original draft.
M Roulet-Matton: validation and investigation.
V Giang: validation and investigation.
R Cardona Hurtado: validation and investigation.
D Chaput: validation and investigation.
S-P Gravel: conceptualization, formal analysis, supervision, funding acquisition, investigation, visualization, methodology, project administration, and writing—original draft, review, and editing.

### Conflict of Interest Statement

The authors declare that they have no conflict of interest.

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
