## [Reviewer comments · Life Science Alliance]

Life Science Alliance

Mitochondrial Signatures Shape Phenotype Switching and Apoptosis in Response to PLK1 Inhibitors

Simon-Pierre Gravel, Émilie Lavallée, Maëline Roulet-Matton, Viviane Giang, Roxana Cardona Hurtado, and Dominic Chaput
DOI: <https://doi.org/10.26508/lsa.202402912>

Corresponding author(s): Simon-Pierre Gravel, Université de Montréal

Review Timeline:

Submission Date:	2024-06-27
Editorial Decision:	2024-08-23
Revision Received:	2024-11-22
Editorial Decision:	2024-11-27
Revision Received:	2024-12-02
Accepted:	2024-12-03

Transaction Report:

August 23, 2024

Re: Life Science Alliance manuscript #LSA-2024-02912-T

Prof. Simon-Pierre Gravel
University of Montreal
Faculty of Pharmacy
2940 chemin de la polytechnique
Pavillon Jean-Coutu, room 3196
Montreal, QC H3T 1J4
Canada

Dear Dr. Gravel,

Thank you for submitting your manuscript entitled "Mitochondrial Signatures Shape Phenotype Switching in Response to PLK1 Inhibition in Melanoma" to Life Science Alliance. The manuscript was assessed by expert reviewers, whose comments are appended to this letter. We invite you to submit a revised manuscript addressing the Reviewer comments.

Thank you for this interesting contribution to Life Science Alliance. We are looking forward to receiving your revised manuscript.

Sincerely,

Eric Sawey, PhD
Executive Editor
Life Science Alliance
<http://www.lsa-journal.org>

B. MANUSCRIPT ORGANIZATION AND FORMATTING:

Reviewer #1 (Comments to the Authors (Required)):

The authors Lavallée et al. present an important work about the role of PLK1 in melanoma cell plasticity directly influencing dedifferentiation and pro-inflammatory responses which are both common therapy escape mechanisms.

Interestingly they provide data, that previous reports dealing with RSK inhibition and RSK isoforms might have seen similar or the same effects due to unspecific inhibition of such inhibitors.

Before publication of the present project a few points should be addressed:

a) major points

1. the paper is difficult to read and the figures contain many data. This is somewhat confusing especially since it is not always clear when the authors have used foreign data for their analysis. Therefore I strongly recommend to clearly mark analysis where foreign datasets were used and clearly name them in the figure legend (e.g. the following publicly available datasets were used in Fig. Xy)

2. The authors mention publications that might contain wrong conclusions due to unspecific effects of RSKi e.g. by Kosnoppfel et al. that could inhibit e.g. PLK1 as well. In the discussion they take up that question and discuss such papers (although not with a focus at the melanoma relevant ones). However, the point about dedifferentiation and pro-inflammation is lacking in the discussion and they mostly discuss effects on proliferation. But I do believe that there are also data about RSK and CXCL8 published. So this needs to be improved with extra literature research including silencing of RSK or downstream genes and CXCL8 or other pro-inflammatory factors.

minor:

1. The introduction contains wrong numbers. Cutaneous melanomas are not BRAF mutated in more than 50% of the patients (if you consider the oncogenic hotspot mutations). Please specify and correct more properly.

2. The final conclusion in the discussion is very blurry. The authors should sharpen this and describe where they might see a gap for the development and usage of new / specific PLK inhibitors or mitochondrial therapies.

Reviewer #2 (Comments to the Authors (Required)):

This study by Émilie Lavallée is a focused analysis of the role of non-specific RSK inhibitors that also inhibit PLK1. The latter inhibitory effect is potentially useful as a therapeutic strategy against melanomas that are molecularly classified as non-inflammatory and lacking mitochondrial functionality to counteract the anticancer activity of the inhibitors.

The experiments are well done and the informatics and statistical analyses seemed to be performed thoroughly using publicly available data and analysis of the study group's own data generated from melanoma cell lines. These results show convincing transcriptional and protein differences among two cell lines that respond differently to RSK and PLK inhibitors.

However, as mentioned below, there are several weaknesses in the study, which stem from choice of experimental models and interpretation of results, which in some instances are an over-interpretation of findings amounting to speculation without support from findings. These overall dampen my enthusiasm for this study.

The most significant weakness is the use of only two melanoma cell lines as models to study the molecular differences of response to RSK/PLK inhibition. While one cell line (MeWO) is BRAF-wild type and has active pigmentation pathway, the other (A375) is BRAF-mutant and does not have an active pigmentation pathway to start with. Both molecular features can influence de-differentiation mechanisms in melanomas, their expression of inflammatory genes like CXCL8 and also their drug responses, in unrelated ways. Additionally the generation of MeWo Rho-0 cells using long-term ethidium bromide treatment would be expected to generate significant mutations that would lead to significant molecular alterations and therapeutic responses unrelated to loss of mitochondrial functionality.

Also the interpretation should be toned down to what the models and the experiments actually show, rather than extrapolating

the findings (for example, in page 15 discussion- "From a clinical perspective, these observations suggest that PLK1 inhibition may quickly lead to resistant satellites associated with immunological remodeling of the tumor microenvironment.", there are no experiments on tumor microenvironment here). The results also do not provide any clear evidence that de-differentiation or phenotype switching might actually be occurring, especially based on results from two very different cell lines tested in vitro.

While I do not suggest a significant overhaul of the study with additional experiments, I will suggest the authors to diligently point out the specific differences in the cell line models used, and the limitations of the study with experimental models and interpretation based on these models.

Reviewer #3 (Comments to the Authors (Required)):

Lavallee et al have investigated the off-target effects of inhibitors of the p90 ribosomal S6 kinase (RSK) family. Briefly, RSK1/2 were shown to be up-regulated in melanoma cells and to act as predominant targets of ERK in melanoma. RSK1/2 were also shown to mediate resistance to chemotherapy and BRAF inhibition. Recent reports have proposed that RSK1/2 are potential therapeutic targets in melanoma with constitutive MAPK activation to overcome resistance to BRAF and MEK inhibitors. However, several RSK inhibitors are known for unspecific effects. For example, the unselective RSK inhibitor BI-1870 was shown to bind to the active site of dozens of unrelated kinases such as Polo-like kinase (PLK) 1/3, aurora kinase B, and PIK3, leading to RSK-independent effects. The authors set out to characterize those unspecific effects in a number of cell lines and found that the anti-proliferative effects of several RSK inhibitors might be caused by off-target inhibition of other protein kinases, such as PLK1, a master regulator of cell division and candidate for targeted therapy with increased expression in melanoma. Their results show that PLK1 is a putative target of unselective RSK inhibitors, mediating their anti-proliferative and pro-inflammatory effects. In addition, Lavallee et al show that the observed heterogeneity between cell lines in terms of the response to RSK and PLK1 inhibition is affected by the expression of mitochondrial proteins associated with a mitochondrial resistance signature. The manuscript is very interesting and of great relevance to the field. The experiments are well conducted and thought through, and the paper is well written. The findings are important and will certainly add to the field of melanoma research and the search for therapeutic targets.

I have two major comments that I feel should be addressed prior to publication.

Major Comments:

1) In Figure 4, the authors examine the effect of PLK1 inhibition/ knock-down on MeWo and A375 cells to compare those to the effect of the unselective RSK inhibitor BI-D1870. Similar to the treatment with the RSK inhibitors the authors observe a pro-inflammatory response in MeWo cells and reduced proliferation. Interestingly, A375 cells undergo apoptosis. While the effects of BI-D1870 (RSK) and BI 6727 (PLK1), as well as PLK1 KD are very comparable in MeWo cells (reduced proliferation and increased CXCL8 levels), it seems they are not completely comparable in A375 cells. While the authors do not find an increase in CXCL8 levels, as shown for BI-D1870, they find an increase in apoptosis of A375 cells after PLK1 inhibition/ knock down. From the manuscript I can not tell if A375 cells undergo apoptosis after treatment with BI-D1870. Could the authors please comment on this difference?

2) To explore the potential role of mitochondria in shaping the pro-inflammatory and resistant phenotype of MeWo cells, the authors generated a mitochondria incompetent MeWo cell line (Rho-0) to quantify their response to PLK1 inhibition (Figure 6). Interestingly, the Rho-0 cells were more sensitive to BI 6727 treatment in comparison to MeWo cells and displayed more apoptosis and less inflammatory response. This is very interesting.

Figure 6 L and M each show the relative viable cell count in response to BI 6727 treatment. Figure M would benefit by adding the data for untreated cells to see if the mitochondria incompetent cells already have lower viability counts than their "healthy" counterparts, even without BI 6727 treatment. Or does "relative viable cell count" refer to a fraction in comparison to untreated cells. This is not clear from the figure, text and material & methods and the authors should clarify if the Rho-0 cell lines have already defects in viability and what "relative viable cell count" refers to.

We thank the reviewers for their constructive comments. Modified sections in the manuscript have been highlighted in yellow to facilitate tracking.

Reviewer 1

R1_Comment 1: the paper is difficult to read and the figures contain many data. This is somewhat confusing especially since it is not always clear when the authors have used foreign data for their analysis. Therefore I strongly recommend to clearly mark analysis where foreign datasets were used and clearly name them in the figure legend (e.g. the following publicly available datasets where used in Fig. Xy)

R1_Answer 1: We agree that analyses derived from publicly available datasets should always be well-identified to segregate them from our own datasets. We made sure that is issue was fixed in all panels where such data are used. As recommended, we included a sentence about publicly available datasets in the figure legends of Figures 1A-B, 6A-C and 7B. We also added this information in supplemental figure legends S1A and S5B.

Here are the changes we made:

- Figure 1. (A) The following publicly available dataset was used: Skin Cutaneous Melanoma (TCGA, PanCancer Atlas).
- Figure 1. (B) The following publicly available dataset was used: DepMap Portal, CRISPR (DepMap Public 24Q2+Score, Chronos) for the indicated genes.
- Figure 6. (A) The following publicly available dataset was used: Cancer Cell Line Encyclopedia (Broad, 2019).
- Figure 6. (B, C) The following publicly available dataset was used: Cancer Cell Line Encyclopedia (Broad, 2019).
- Figure 7 (B) The following publicly available datasets were used: Cancer Cell Line Encyclopedia (Broad, 2019); DepMap Portal: PLK1 CRISPR (DepMap Public 24Q2+Score, Chronos), PLK1 RNAi (Achilles+DRIVE+Marcotte, DEMETER2), Drug sensitivity data for the PLK1 inhibitors BI 6727, BI-2536, GSK-461364, HMN-214, NMS-1286937, GW-843682X (PRISM repurposing primary screen).
- Figure S1. (A) The following publicly available dataset was used: DepMap Portal, RNAi (Achilles+DRIVE+Marcotte, DEMETER2) for the indicated genes.

- Figure S5. (B) (...) using RNA-Seq data from a publicly available dataset: Skin Cutaneous Melanoma (TCGA, PanCancer Atlas).

R1_Comment 2: The authors mention publications that might contain wrong conclusions due to unspecific effects of RSKi e.g. by Kosnoppfel et al. that could inhibit e.g. PLK1 as well. In the discussion they take up that question and discuss such papers (although not with a focus at the melanoma relevant ones). However, the point about dedifferentiation and pro-inflammation is lacking in the discussion and they mostly discuss effects on proliferation. But I do believe that there are also data about RSK and CXCL8 published. So this needs to be improved with extra literature research including silencing of RSK or downstream genes and CXCL8 or other pro-inflammatory factors.

R1_Answer 2: We thank the reviewer for his comment. We initially focused on other topics in the conclusions due to space limitations. Based on the comments of the 3 reviewers, we expanded the discussion by developing relevant topics such as dedifferentiation, mutational status and pro-inflammatory effects. This section can be found on page 18. To keep the focus on our main findings, we decided to not insist on papers with possibly wrong conclusions about the role of RSK in melanoma. This has been the focus of previous in-depth articles on the off-target effects of unselective inhibitors such as BI-D1870 that we have already cited. Discussion on RSK was limited to the section at the end of page 17.

R1_Comment 3: The introduction contains wrong numbers. Cutaneous melanomas are not BRAF mutated in more than 50% of the patients (if you consider the oncogenic hotspot mutations). Please specify and correct more properly.

R1_Answer 3: We thank the reviewer for his comment. This was a mistake. We meant that BRAF mutations are found in approximately 40-60% of melanoma patients, not in more than 40-60%. However, a meta-analysis of 32 published articles (6299 patients, see reference below) found 3 studies that show an incidence of > 60% for BRAF mutations in the studied population. Since we did not specify hotspot mutations in this sentence, we fixed the original sentence with the word <<approximately >> and included this new supporting reference in the introduction on page 3:

- Gutierrez-Castaneda LD, Nova JA, Tovar-Parra JD (2020) Frequency of mutations in BRAF, NRAS, and KIT in different populations and histological subtypes of melanoma: a systemic review. *Melanoma Res* 30: 62-70

R1_Comment 4: The final conclusion in the discussion is very blurry. The authors should sharpen this and describe where they might see a gap for the development and usage of new / specific PLK inhibitors or mitochondrial therapie.

R1_Answer 4: We agree with the reviewer that the final conclusion was rather short (495 characters) and unfocused. In line with the changes we have made in the discussion, the final conclusion was fully rewritten and expanded to better reflect the state of research and the challenges to improve the usage of PLK1 inhibitors. The final conclusion can be found on page 20.

Reviewer 2

R2_Comment 1: The most significant weakness is the use of only two melanoma cell lines as models to study the molecular differences of response to RSK/PLK inhibition. While one cell line (MeWo) is BRAF-wild type and has active pigmentation pathway, the other (A375) is BRAF-mutant and does not have an active pigmentation pathway to start with. Both molecular features can influence de-differentiation mechanisms in melanomas, their expression of inflammatory genes like CXCL8 and also their drug responses, in unrelated ways.

R2_Answer 1: We thank the reviewer for his comment. We now recognize this limitation in the discussion. We include a section on the differences between MeWo and A-375 cell lines and discuss factors that could explain the observed responses, including driver mutations (BRAF V600, NF1 loss) and MITF/PGC-1 α expression status (linked to pigmentation pathways). The section starts on page 18.

R2_Comment 2: Additionally the generation of MeWo Rho-0 cells using long-term ethidium bromide treatment would be expected to generate significant mutations that would lead to significant molecular alterations and therapeutic responses unrelated to loss of mitochondrial functionality.

R2_Answer 2: We agree with the reviewer that the Rho-0 model has limitations including the introduction of mutations. We now include a section on the Rho-0 model and discuss viability and mutations. We also suggest an alternative approach such as the targeting of TFAM, a major mitochondrial biogenesis factor. This addition can be found in the middle of page 17.

R2_Comment 3: Also the interpretation should be toned down to what the models and the experiments actually show, rather than extrapolating the findings (for example, in page 15 discussion- "From a clinical perspective, these observations suggest that PLK1 inhibition may quickly lead to resistant satellites associated with immunological remodeling of the tumor microenvironment.", there are no experiments on tumor microenvironment here).

R2_Answer 3: We thank the reviewer for pointing this out. We removed this sentence from the discussion.

R2_Comment 4: The results also do not provide any clear evidence that de-differentiation or phenotype switching might actually be occurring, especially based on results from two very different cell lines tested in vitro.

R2_Answer 4: We thank the reviewer for this comment. Although we provide transcriptomic data (Figures 2J and 5F), qRT-PCR data (Figure 5A-C, E, G-H), cytokine array data (Figures 2H,I and S2D-E) and immunoblots (Figures 2K,L and 5D) that support key aspects of phenotype switching, we now recognize the limitation of using only two cell lines in the discussion. As mentioned above, we now discuss the factors that could influence the response to PLK1 inhibition in A375 and MeWo cell lines, such as driver mutations and MITF/PGC-1 α status. We also mention in the final conclusion that the de-differentiating effects of PLK1 targeting will have to be investigated in a larger number of cell lines. This can be found on pages 18-20.

R2_Comment 5: While I do not suggest a significant overhaul of the study with additional experiments, I will suggest the authors to diligently point out the specific differences in the cell line models used, and the limitations of the study with experimental models and interpretation based on these models.

R2_Answer 5: As indicated in Answer 1, we included an articulated discussion on the differences between MeWo and A-375 cell lines and discussed the limitations of the Rho-0 model. Overall, we believe that the discussion and conclusion were more focused on interpretation, limitations and next challenges.

Reviewer 3

R3_Comment 1: In Figure 4, the authors examine the effect of PLK1 inhibition/ knock-down on MeWo and A375 cells to compare those to the effect of the unselective RSK inhibitor BI-D1870. Similar to the treatment with the RSK inhibitors the authors observe a pro-inflammatory response in MeWo cells and reduced proliferation. Interestingly, A375 cells undergo apoptosis. While the effects of BI-D1870 (RSK) and BI 6727 (PLK1), as well as PLK1 KD are very comparable in MeWo cells (reduced proliferation and increased CXCL8 levels), it seems they are not completely comparable in A375 cells. While the authors do not find an increase in CXCL8 levels, as shown for BI-D1870, they find an increase in apoptosis of A375 cells after PLK1 inhibition/ knock down. From the manuscript I can not tell if A375 cells undergo apoptosis after treatment with BI-D1870. Could the authors please comment on this difference?

R3_Answer 1: We did not investigate the pro-apoptotic effects of BI-D1870 (RSK/PLK inhibitor) in A-375, as we did not focus on RSK inhibition after Figure 3. However, BI-D1870 has anti-inflammatory effects in A-375 cells (Figure 1D-F), which is not observed when these cells are treated with BI6727 (PLK1 inhibitor) or with siRNA-PLK1 (Figure 4A and 4I). Since specific

RSK inhibition with LJM685, BIX02565, and LJM308 has anti-inflammatory effects in A-375 cells (Figure 1E and 1F), we believe that the difference between BI-D1870 and BI 6727 could be attributed to RSK inhibition. It is likely that RSK inhibition reduces the bioenergetic charge and associated oxidative stress in A-375 (as mentioned in the discussion) and that these cells become less sensitive to PLK1 inhibition and, thus less apoptotic. This relationship between RSK and PLK1 inhibition could be investigated in the future, and we thank the reviewer for this very interesting observation.

R3_Comment 2: Figure M would benefit by adding the data for untreated cells to see if the mitochondria incompetent cells already have lower viability counts than their "healthy" counterparts, even without BI 6727 treatment. Or does "relative viable cell count" refer to a fraction in comparison to untreated cells. This is not clear from the figure, text and material & methods and the authors should clarify if the Rho-0 cell lines have already defects in viability and what "relative viable cell count" refers to.

R3_Answer 2: We are glad to clarify this point. Labels of viability graphs were mislabeled. We now indicate in our five viability graphs that data are expressed as % of maximal viability (y-axis), which is what a PrestoBlue HS viability assay can provide. This is now detailed in **Materials and methods (page 23)**, and **figure legends of Figures 4B (page 46), 6L (page 48), 7I (page 49), S1C S1D, and Results (page 9)** have been fixed. Figure 6M shows only a 10 μ M BI 6727 dose while a range of doses is shown in 6L. In 6L, low doses of BI6727 (0.01, 0.1 and 1 nM) have no impact on viability and are set at 100% viability on average. In these experiments, we are examining the fold change versus 100% viability, with each cell line being tested independently. These graphs do not show how viability is different between parental MeWo and Rho-0 cells at baseline. However, we now include a sentence in the **Material and methods (page 21)** and the **discussion (page 17)** in which we indicate a reduction of ~14% viability at baseline in Rho-0 cells versus MeWo, as tested by trypan blue exclusion assay. If this small drop in viability had been considered into 6L or 6M, the viability of Rho-0 cells would have been lower at all doses. Thus, this modification would not have changed our interpretation of results and would not have impacted IC50s.

November 27, 2024

RE: Life Science Alliance Manuscript #LSA-2024-02912-TR

Prof. Simon-Pierre Gravel
Université de Montréal
Faculty of pharmacy
2940 chemin de la polytechnique
Pavillon Jean-Coutu, room 3196
Montreal, QC H3T 1J4
Canada

Dear Dr. Gravel,

Thank you for submitting your revised manuscript entitled "Mitochondrial Signatures Shape Phenotype Switching and Apoptosis in Response to PLK1 Inhibitors". We would be happy to publish your paper in Life Science Alliance pending final revisions necessary to meet our formatting guidelines.

- please be sure that the authorship listing and order is correct
- please upload your manuscript text as an editable doc file
- please upload both your main and supplementary figures as single files
- please add your supplementary figure legends to the main manuscript text

A. FINAL FILES:

B. MANUSCRIPT ORGANIZATION AND FORMATTING:

Thank you for your attention to these final processing requirements. Please revise and format the manuscript and upload materials within 5 days.

Sincerely,

December 3, 2024

RE: Life Science Alliance Manuscript #LSA-2024-02912-TRR

Prof. Simon-Pierre Gravel
Université de Montréal
Faculty of pharmacy
2940 chemin de la polytechnique
Pavillon Jean-Coutu, room 3196
Montreal, QC H3T 1J4
Canada

Dear Dr. Gravel,

Thank you for submitting your Research Article entitled "Mitochondrial Signatures Shape Phenotype Switching and Apoptosis in Response to PLK1 Inhibitors". It is a pleasure to let you know that your manuscript is now accepted for publication in Life Science Alliance. Congratulations on this interesting work.

DISTRIBUTION OF MATERIALS:

Again, congratulations on a very nice paper. I hope you found the review process to be constructive and are pleased with how the manuscript was handled editorially. We look forward to future exciting submissions from your lab.

Sincerely,
